# Environmental cues from neural crest derivatives act as metastatic triggers in an embryonic neuroblastoma model

Dounia Ben Amar[1], Karine Thoinet[1], Benjamin Villalard [1], Olivier Imbaud[1], Clélia Costechareyre[2], Loraine Jarrosson[2], Florie Reynaud[1], Julia Novion Ducassou[3], Yohann Couté [3], Jean-François Brunet[4], Valérie Combaret[5], Nadège Corradini[6], Céline Delloye-Bourgeois [1,7✉] & Valérie Castellani [1,7✉]

Embryonic malignant transformation is concomitant to organogenesis, often affecting multipotent and migratory progenitors. While lineage relationships between malignant cells and their physiological counterparts are extensively investigated, the contribution of exogenous embryonic signals is not fully known. Neuroblastoma (NB) is a childhood malignancy of the peripheral nervous system arising from the embryonic trunk neural crest (NC) and characterized by heterogeneous and interconvertible tumor cell identities. Here, using experimental models mimicking the embryonic context coupled to proteomic and transcriptomic analyses, we show that signals released by embryonic sympathetic ganglia, including Olfactomedin-1, induce NB cells to shift from a noradrenergic to mesenchymal identity, and to activate a gene program promoting NB metastatic onset and dissemination. From this gene program, we extract a core signature specifically shared by metastatic cancers with NC origin. This reveals non-cell autonomous embryonic contributions regulating the plasticity of NB identities and setting pro-dissemination gene programs common to NC-derived cancers.

[1] University of Lyon, University Claude Bernard Lyon 1, MeLiS, CNRS UMR5284, INSERM U1314, NeuroMyoGene Institute, 69008 Lyon, France 8 avenue Rockefeller. [2] Oncofactory SAS, 8 avenue Rockefeller, 69008 Lyon, France. [3] University Grenoble Alpes, INSERM, CEA, UMR BioSanté U1292, CNRS, CEA, FR2048 38000, Grenoble, France. [4] Institut de Biologie de l'ENS (IBENS), Inserm, CNRS, École normale supérieure, PSL Research University, Paris, France. [5] Laboratory of Translational Research, Léon Bérard Centre, Lyon, France. [6] Departments of Oncology and Clinical Research, Centre Léon Berard and Institut d'Hématologie et d'Oncologie Pédiatrique, Lyon, France. [7] These authors jointly supervised this work: Céline Delloye-Bourgeois, Valérie Castellani. ✉email: celine.delloye@univ-lyon1.fr; valerie.castellani@univ-lyon1.fr

Half of childhood malignancies have an embryonic origin, with often highly disseminated aggressive forms already established at the diagnosis. The lack of access to initial tumorigenic and metastatic processes considerably limits the understanding of their etiology[1,2]. While the lineage relationships of malignant cells with their potential cells of origin is the matter of extensive investigations[3–6], it is still unknown whether exogenous signaling of the embryonic organogenesis, acting non-cell autonomously to regulate cell proliferation and migration, also account for setting clinical characteristics of childhood malignancies.

Neuroblastoma has an early embryonic origin and primarily affects sympathoadrenal tissues. The sympathoadrenal lineage derived from the neural crest (NC) and Schwann cell precursors (SCPs) are major embryonic progenitor populations at the origin of these organs. Their morphogenesis starts at early embryonic stages[7–10] and is orchestrated by multiple extracellular signaling emanating from diverse sources acting at distance or locally, regulating the directed migration, the proliferation, the coalescence of the NC and the differentiation of its derivatives[11]. Malignant transformation of cells of the sympathoadrenal NC lineage (SA NCCs) and potentially of SCPs gives rise to neuroblastoma[12,13]. This malignancy predominantly affects children in early infancy, and can be manifested even before birth. The exact etiology of NB remains debated and under intense investigation. A hallmark of NB is the high heterogeneity of the tumors, with co-existence of cells with different identities and phenotypical conversions, as revealed by several recent transcriptional analyses[3]. Likewise, two major cell identities defined by transcriptional programs and controlled by specific core regulatory circuits have been recently uncovered: NCC-like mesenchymal (MES) and noradrenergic (NOR) identities[14,15]. NB heterogeneity is also reflected in the metastatic disease, with highly disseminated forms that can either spontaneously regress or lead to fatal outcome. The bases of NB heterogeneity are fully unclear. They may take their roots in the phenotypic plasticity acquired by malignant cells, but also in the embryonic context and the nature of the cell of origin, the NC. It is a transitory lineage, restricted to the embryonic stage, which holds unique properties in chordates, combining a high degree of multipotency and a remarkable arsenal of migratory strategies that evolve together with physical features of growing tissues[12,13,16].

The clinical presentation of NB metastatic disease is a major hurdle limiting our understanding of metastatic NB etiology. On the one hand, the molecular environment of NB cells preparing for metastasis remains inscrutable in patients. On the other hand, animal models recapitulating neuroblastoma disease in an embryonic host lack, and most of them do not reproduce the metastatic disease[17–19]. Whether cellular and molecular features revealed by anatomo-pathology studies of tumor samples[20] reflect the environmental context at the time when NB cells engaged towards secondary dissemination is unlikely. The metastatic onset could occur very early and be influenced by environmental factors regulating organogenesis.

Initially discovered in the olfactory neuroepithelium, Olfactomedin1 (OLFM1) protein, also referred to as Noelin1 or Pancortin, belongs to a vast and conserved super family of secreted glycoproteins, that shares a 250 amino-acid olfactomedin domain mediating protein-protein interactions. Interestingly in the chicken embryo, high levels of *OLFM1* transcripts were detected by in situ hybridization in the NC and developing sympathetic chains, [http://geisha.arizona.edu/geisha/search.jsp?entrez_gene=449625]. In E14 and E17 mouse embryo, OLFM1 expression was also reported in the adrenal gland [https://www.ebi.ac.uk/gxa/genes/ensmusg00000026833]. Moreover, manipulations of OLFM1 levels in the chicken embryo were found to alter the generation of NCCs[21]. More generally, various contributions of OLFM family members were reported in the developing nervous system[22–24], ie- in the regulation of optic nerve formation[25], the generation of Hippocampal GABAergic interneurons[26], nodes of Ranvier formation[27] and migration of cortical neurons[28]. Although OLFM1 has not so far been linked to NB, contributions of OLFM family members to cancer have been largely documented[29–32]. OLFM proteins are multifunctional, regulating cell adhesion, proliferation, differentiation and apoptosis[33]. These versatile properties might rely on the complex networks of signaling molecules functionally linked to OLFM proteins and their context-dependent regulations. Very little is yet known on their mechanisms of action although their activity has been correlated not only to several receptors, such as amyloid precursor protein (APP)[34], NOGOR/NGR/RTN4R, and AMPAR/GRIA2 receptors[23,35,36] but also to the modulation of the WNT pathway. Several signaling molecules are thought to act as OLFM-related effectors in specific contexts, including Bcl-XL, WAVE-1, DISC1 and β-dystrobrevin[23,37–39]. Hence, OLFM proteins are expected to exert a range of effects depending on the molecular environment in which they are recruited.

Here we investigate whether physiological signals from the embryonic development could modulate the behavior of NB cells. We show that NC-derived peripheral ganglia -the dorsal root ganglia (DRG) and the sympathetic ganglia- produce exogenous factors including OLFM1 that set a specific transcriptomic program in NB cells, promoting their detachment from the primary tumor and their metastatic dissemination.

## Results

### SG^cm impacts on NB cell–cell cohesion and favors NB migration and invasion

To assess signals regulating embryonic sympathetic morphogenesis, we first set up a technique to produce media conditioned by nascent sympathetic ganglia (SG^cm). We harvested sympathetic chains from series of E6 to E10 chick embryos, and cultured them for 48 h (Fig. 1a). Quality and homogeneity of SG^cm concentrations between conditions were checked by detecting tyrosine hydroxylase protein expression in Western blot (Supplementary Fig. 1a). We then exposed human NB cells to these conditioned media, using a hanging drop procedure inducing cells to form an adhesive aggregate that mimics a tumor, as previously described[40]. We found that E6 to E9-cSG^cm significantly interfered with NB IGR-N91 and SH-SY5Y aggregate formation, normally achieved within 24 h of culture, (Fig. 1b, c and Supplementary Fig. 1b, c). This inhibitory effect was developmentally down-regulated and lost with E10-cSG^cm suggesting the implication of transiently expressed sympathetic cues. Paraformaldehyde fixation of sympathetic tissues prior to the culture abrogated the inhibitory effect, validating the implication of cues actively released by SGs (Fig. 1d, e). Interestingly, similar inhibition of cell–cell cohesion was observed with SG^cm produced from E15.5 mouse embryo sympathetic chains, a stage equivalent to E8.5 in the chick embryo according to Carnegie stages classification[41,42] thus indicating a conserved mechanism in mammals (Fig. 1f, g). We next tested whether SG^cm-mediated downregulation of cell-cell cohesion had an impact on NB migratory and/or invasive properties. We performed transwell assays with IGR-N91 and SH-SY5Y NB cell lines plated on porous membranes -coated or not with Matrigel- and applied E6 or E8-cSG^cm in lower chambers (Fig. 1h and Supplementary Fig. 1d–f). E6-cSG^cm and even more effectively E8-cSG^cm increased NB cell migration and invasion. As paravertebral neuroblastomas emerge in a complex environment that is not solely composed of sympathetic chains, we wondered whether other closely affixed NC-derived structures, such as the dorsal

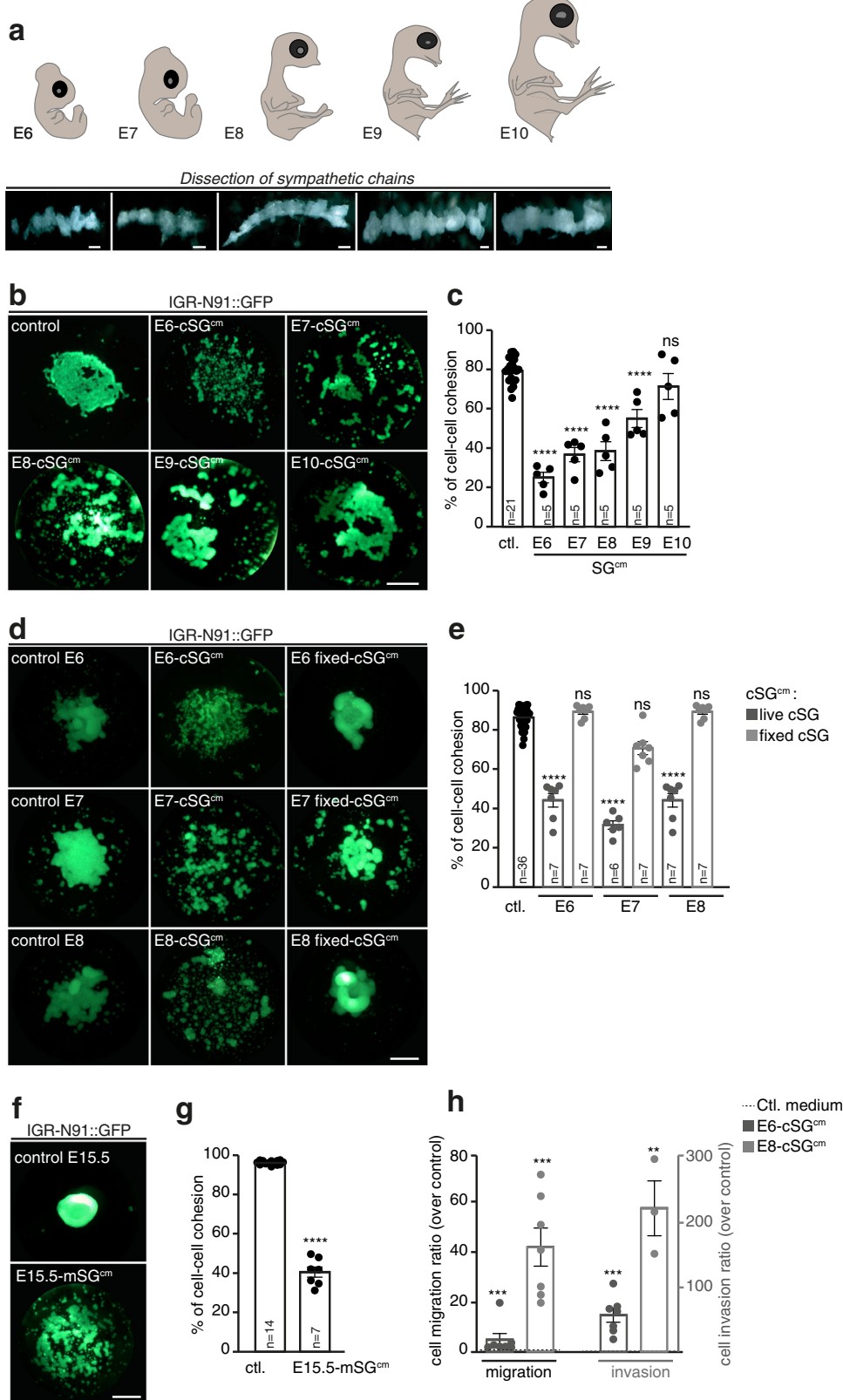

root ganglia (DRG), could also release pro-dissemination signals for NB cells. We found that cDRG^cm produced from E6 to E8 embryos and mDRG^cm from E15.5 embryos exerted inhibitory effects on NB cell-cell cohesion (Supplementary Fig. 2a–e) and migration/invasion (Supplementary Fig. 2f–h)

with a developmental drop similar to that of SG^cm. This suggested that signals of peripheral nerve ganglia organogenesis act as pro-metastatic triggers for NB cells, by downregulating cell-cell cohesion in the primary tumor, promoting NB cell detachment and dissemination onset.

**Fig. 1 Sympathetic Ganglia conditioned medium (SGᶜᵐ) induces a decrease in neuroblastoma cell-cell cohesion. a** Illustration of the dissection procedure of chick embryonic sympathetic chains from E6 to E10 developmental stages. Upper panels are schematic representations of chick embryos and lower panels show representative pictures of dissected sympathetic chains. Scale bar: 1 mm. **b, c** Representative pictures of IGR-N91 cell aggregates (**b**) and quantification (**c**) of cell-cell aggregation rate of IGR-N91 cells cultured in hanging drops and treated with E6 to E10-cSGᶜᵐ, compared to medium without any cultured tissue (ctl.) ($N = 5$ independent experiments; n: number of aggregates analyzed per condition; two-sided Mann–Whitney $U$ test; comparison to control medium: E6- to E9-cSGᶜᵐ: $p < 0.0001$, E10: $p = 0.3476$). Scale bar: 1 mm. **d, e** Representative pictures of IGR-N91 cell aggregates (**d**) and quantification (**e**) of of cell-cell aggregation rate of IGR-N91 cells cultured in hanging drops and treated with E6 to E8-cSGᶜᵐ, prepared with live or fixed sympathetic ganglia compared to medium without any cultured tissue (ctl.) ($N = 6$ independent experiments; n: number of aggregates analyzed per condition; two-sided Mann–Whitney $U$ test; comparison to control medium: E6-, E7-, E8-cSGᶜᵐ: $p < 0.0001$, E6 fixed-cSGᶜᵐ: $p = 0.2070$, E7 fixed-cSGᶜᵐ: $p = 0.0012$, E8 fixed-cSGᶜᵐ: $p = 0.2070$). Scale bar: 1 mm. **f, g** Representative pictures of IGR-N91 cell aggregates (**f**) and quantification (**g**) of cell-cell aggregation rate of IGR-N91 cells cultured in hanging drops and treated with E15.5-mSGᶜᵐ, compared to medium without any cultured tissue (ctl.) ($N = 7$ independent experiments; n: number of aggregates analyzed per condition; two-sided Mann–Whitney $U$ test; comparison to control medium: $p < 0.0001$). Scale bar: 1 mm. **h** Quantification of IGR-N91 cells migration and invasion properties in transwell assays using E6 and E8-cSGᶜᵐ in the lower part of the device. Ratios over the number of migrating/invading cells in the control condition (medium without any cultured tissue in the lower part) are shown. ($N = 7$ independent experiments with E6-cSGᶜᵐ, $N = 3$ independent experiments with E8-cSGᶜᵐ; two-sided Mann–Whitney $U$ test; comparison to control medium, migration/invasion: E6-cSGᶜᵐ: $p = 0.0006/p = 0.0006$; E8-cSGᶜᵐ: $p = 0.0008/p = 0.0083$). Error bars show SEM. Source data are provided as a Source Data file.

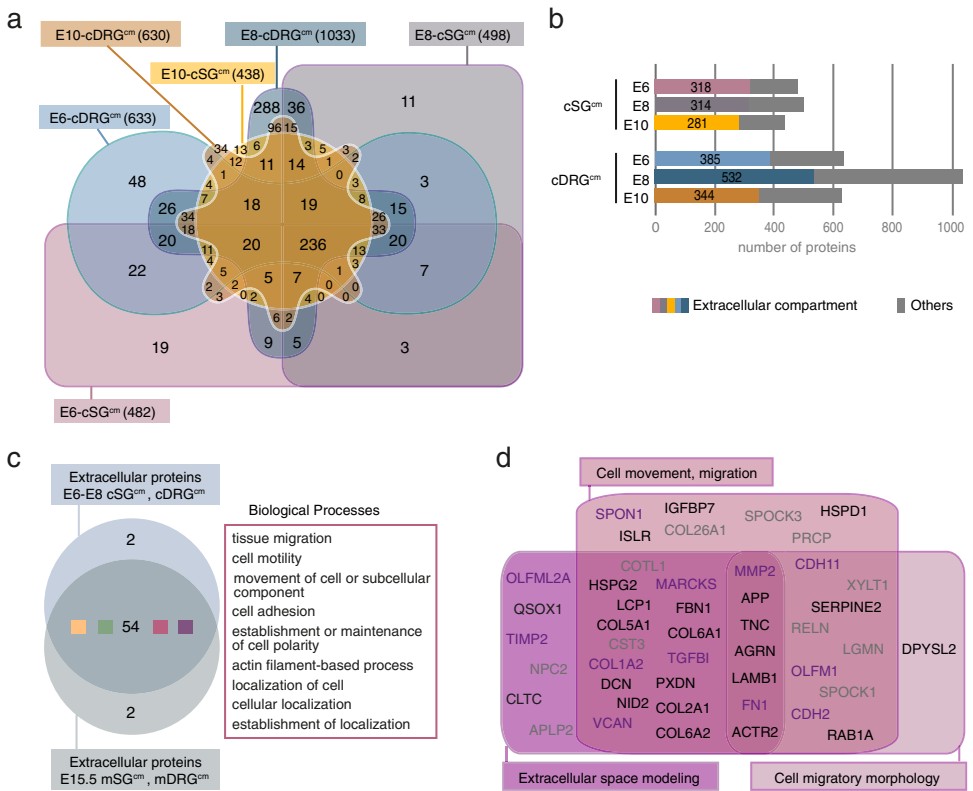

**Fig. 2 Sympathetic Ganglia conditioned medium (SGᶜᵐ) contains a set of proteins involved in cell motility. a** Venn diagram indicating the number of proteins detected in the 6 types of chick conditioned media: E6-, E8- and E10-cSGᶜᵐ and E6-, E8- and E10-cDRGᶜᵐ[71]. Samples were analyzed from 2 independent experiments. **b** Fraction of proteins depicted to have an extracellular localization in each chick conditioned medium. **c** Venn diagram performed on biological processes items significantly represented in extracellular proteins present in E15.5-mSGᶜᵐ and in cSGᶜᵐ and cDRGᶜᵐ at E6 and E8. Common biological processes related to cell movement and migration are outlined in pink. **d** Candidate extracellular proteins present in conditioned media triggering NB loss of cell-cell cohesion and extracted from biological processes related to cell motility (items in pink in Supplementary Fig. 3b). In black and purple: proteins or related proteins found in both chick and mouse decohesive conditioned media. In purple, proteins reported in the literature to be involved in neural crest-related processes (see references in Supplementary Fig. 3c).

**Proteomic characterization of SGᶜᵐ and DRGᶜᵐ highlights developmental cues involved in cell movement.** To identify such microenvironmental triggers, we next characterized by mass spectrometry the proteomic content of the conditioned media (Fig. 2a). We extracted a set of 236 proteins detected in conditioned media from both cSGᶜᵐ and cDRGᶜᵐ. We found 218 proteins classified as extracellular in both E6 to E8 SGᶜᵐ and

DRGᶜᵐ (Fig. 2b). Four major ontology classes of biological processes significantly emerged, with proteins involved in "cell movement and migration", "Response to extracellular cues and signaling", "Metabolism" and "Immune signaling" (Supplementary Fig. 3a and Supplementary Table 1). Moreover, among 58 significantly represented biological processes, 54 were commonly found in E15.5-mSGᶜᵐ/mDRGᶜᵐ, with enrichment in

processes related to cell movement and migration (Fig. 2c and Supplementary Table 2). 108 extracellular proteins were detected in conditioned media having an effect on NB cells (E6-E8 cDRG<sup>cm</sup> and cSG<sup>cm</sup>) and not or barely detected in the inactive ones (E10 cDRG<sup>cm</sup> and cSG<sup>cm</sup>) (Supplementary Fig. 3b). They were mainly associated with biological processes related to extracellular space modeling, cell movement and migration, and cell migratory morphology. We then extracted a list of 35 proteins with related or homologous items also identified in E15.5 mSG<sup>cm</sup> and mDRG<sup>cm</sup> proteomes (Fig. 2d). Notably, 12 proteins had already been associated in the literature with the regulation of neural crest-related processes (Fig. 2d and Supplementary Fig. 3c), among which two members of the olfactomedin family, namely OLFM1 and OLFML2A.

**Cell adhesion and migratory programs are modified in NB cells exposed to embryonic sympathetic cues.** Next, we characterized the impact of microenvironmental cues on the transcriptome of NB cells. IGR-N91 cell aggregates were harvested after 24 h exposure to E8-cSG<sup>cm</sup>, E8-cDRG<sup>cm</sup> or E15.5-mSG<sup>cm</sup> for bulk RNASeq analyses (Fig. 3a). For each condition, the number of significantly upregulated and downregulated transcripts compared to control medium were homogeneously distributed (Fig. 3a). We extracted 308 transcripts showing similar significant variations in all three active conditioned media compared to control neutral medium (Fig. 3b). Two major interconvertible and epigenetically-driven identities have been outlined across NB cell lines and patient samples: a noradrenergic (NOR) identity and a mesenchymal (MES) identity showing close proximity with that of the neural crest, and suspected to carry neuroblastoma resistance to therapeutic agents[14,15]. To first characterize yet unreported IGR-N91 cell identity, key NOR and MES genes expression was compared to that of well-characterized mesenchymal (SHEP) and noradrenergic (SH-SY5Y) NB cell lines[14] (Supplementary Fig. 4a, b). Protein and mRNA expressions of PHOX2B, GATA3, HAND1, HAND2 but barely or not of JUN, RUNX1 or MAML2 were unambiguously in favor of an SH-SY5Y-like noradrenergic identity of IGR-N91 cells. Remarkably, exposure of IGR-N91 aggregates to SG<sup>cm</sup>/DRG<sup>cm</sup> triggered a global and significant downregulation of the NOR signature, that was mirrored by an increased expression of the gene set associated with the MES identity (Fig. 3c and Supplementary Fig. 4c)[14,15]. Such a NOR-to-MES shift was confirmed at the protein level by western blot analysis of two NB cell lines – IGR-N91 and SH-SY5Y- exposed to E8-cSG<sup>cm</sup> (Supplementary Fig. 4d). We previously reported the implantation of IGR-N91 cells in the chick embryo model that formed tumors in the developing SG[40]. Interestingly, comparative RNAseq analysis of pre-grafted cells and tumors formed in SG also showed such a transition towards a MES identity (Supplementary Fig. 4e, f).

Next, we further assessed the functional properties of NB cell aggregate acquired upon exposure to embryonic cues (Fig. 3d). Analyses of Gene Set Enrichment (GSEA) showed highly significant modulations of gene signatures related to cell-cell adhesion, cell-matrix contact, and cell migration in conditioned media conditions compared to control (Fig. 3d, e and Supplementary Fig. 4g, h).

We then conducted cross analyses of transcriptomic and proteomic data sets, focusing on OLFM proteins (Fig. 2d). We generated an "OLFM-related gene set" that integrates major OLFM members, their described receptors and direct interacting/signaling partners (Fig. 3f and Supplementary Table 3). By analyzing this gene set in our RNAseq data, we found a striking clustering of conditioned media-treated NB aggregates apart from control-treated aggregates. Interestingly, these variations

translated into significant enrichment scores of the whole OLFM-related gene set (Supplementary Fig. 4i) for all conditioned media that we found to impact on NB cell behaviors. This suggested that this gene set may contribute to modulate functional properties acquired by NB cells exposed to signals from the paraspinal ganglia.

**OLFM1 boosts NB metastatic properties by triggering NB cell escape from primary tumors.** We then assessed whether microenvironmental OLFM proteins, and more specifically OLFM1, could promote NB cell escape from primary tumors. Consistent with the proteomic data, OLFM1 immunolabeling of E5.5 chick embryos transverse sections was typically high in both DRGs and SGs (Supplementary Fig. 5a). Next, we assessed the effect of recombinant OLFM1 (rOLFM1) on NB cell-cell cohesive properties using our ex vivo aggregation assay (Fig. 4a and Supplementary Fig. 5b). Exposure of NB aggregates to increasing doses of rOLFM1 triggered significant and dose-dependent loss of cell-cell cohesion. Similarly, in transwell migration and invasion assays, increasing doses of rOLFM1 added in the lower chamber promoted NB cells motility, although to a lower extent than complete SG<sup>cm</sup> or DRG<sup>cm</sup> (Fig. 4b and Supplementary Fig. 5c). To assess OLFM1 contribution to cSG<sup>cm</sup> and cDRG<sup>cm</sup> effects, we took advantage of an anti-OLFM1 antibody with reported function-blocking activity in ex vivo set up[24]. Addition of anti-OLFM1 antibody to E8-cSG<sup>cm</sup> and E8-cDRG<sup>cm</sup> almost completely abolished their effect on NB cell-cell cohesion and motility/invasion properties, in contrast to a control isotypic antibody (Fig. 4c, d and Supplementary Fig. 5d–f). This suggested that OLFM1 might functionally be involved in NB cells shift towards a mesenchymal migratory and invasive phenotype. Consistently, adding either recombinant OLFM1 or E8-cSG<sup>cm</sup> to both IGR-N91 and SH-SY5Y cell lines triggered an increased expression of the MES-related genes JUN, MAML2 and RUNX1. This effect of E8-cSG<sup>cm</sup> was, at least partially, lost by adding OLFM1 blocking antibody (Supplementary Fig. 5g).

We next studied whether blocking OLFM1 within NB primary tumor microenvironment could impact on NB propensity to metastasize. We took advantage of our previously set-up in vivo avian model of NB that consists in grafting fluorescent human NB cells within trunk neural crest cells of stage HH14 chicken embryos (E2). Grafted NB cells co-migrate with their physiological counterparts towards sympathoadrenal derivatives in which they first form tumoral masses in 48 h. Secondly, a shift in NB cells gene expression program leads them to detach and engage in dissemination towards distal organs[40]. We engrafted series of embryos with IGR-N91::GFP cells, and performed intravenous injection of OLFM1 Ab 30 h post-grafting (at E3.5), a time when NB cells reach sympathoadrenal derivatives (Fig. 5a). Embryos were collected and imaged with light sheet confocal microscopy at E5 for analysis of primary tumor escape and at E9 for analysis of subsequent distant metastases. Differences between control and OLFM1 Ab-treated groups were already visible at E5. While multiple buds of NB cells escaping from primary tumors were observed in control embryos, primary tumors in OLFM1 Ab-treated embryos were dense, with less buds, evocative of increased cell-cell cohesiveness and decreased potential of dissemination initiation (Fig. 5b, c). Interestingly, the mitotic rate of tumor cells quantified by PH3 immunolabeling was strictly similar in control and OLFM1 Ab-treated groups, showing that NB cell proliferation was unaffected by OLFM1 Ab treatment (Fig. 5d, e). At E9, 3D-imaging of whole embryos revealed that OLFM1 blockade resulted in a significant reduction of the number of metastatic foci and of their distance from primary tumors (Fig. 5f–h). While the mean volume of metastatic foci was higher upon OLFM1

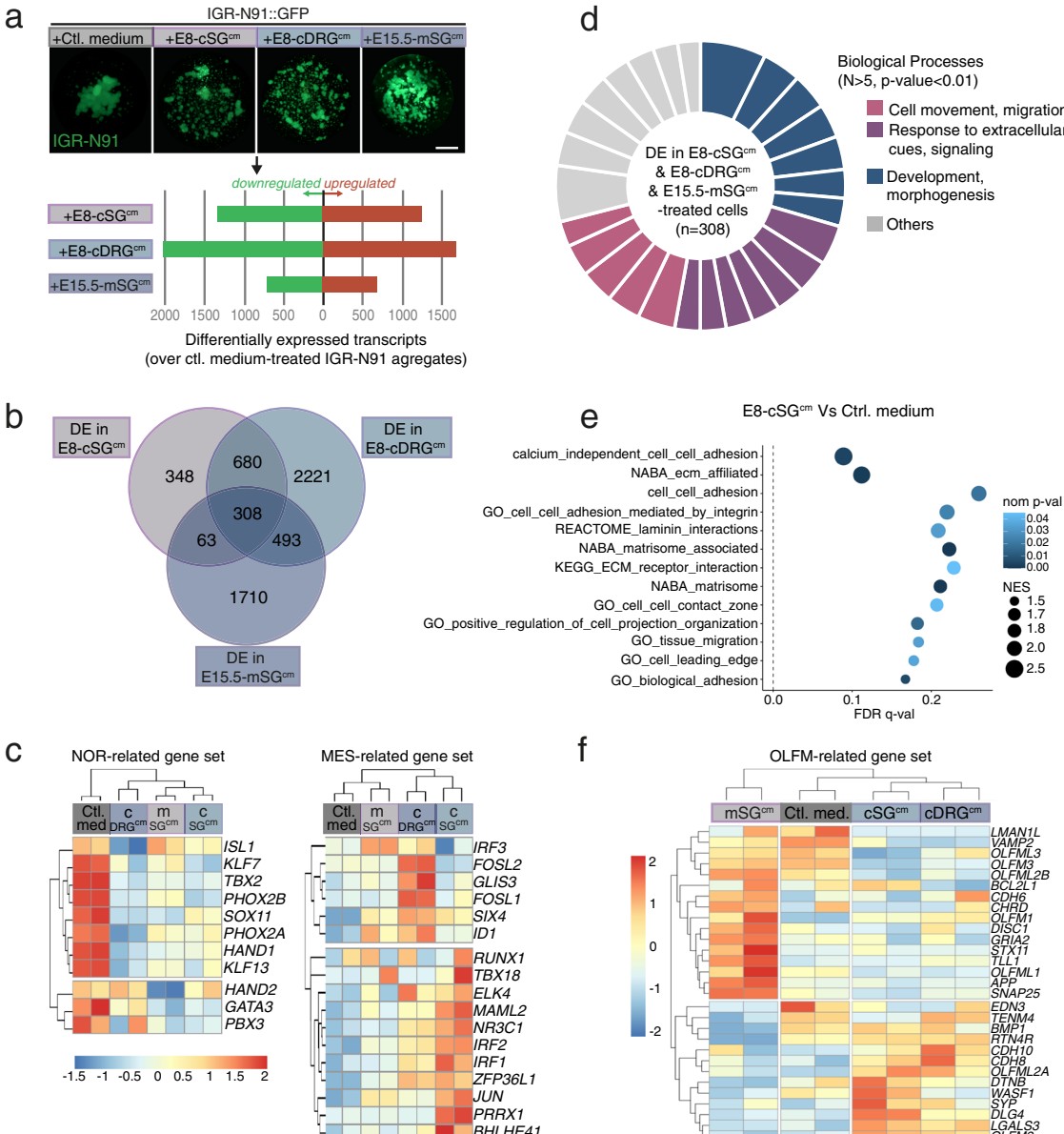

**Fig. 3 Paraspinal secreted cues trigger NB cells shift in gene programs involved in cell–cell cohesion and motility. a** RNASeq analysis performed on IGR-N91 cell cultured in hanging drops and treated with E8-cSGcm, E8-cDRGcm or E15.5-mSGcm (upper panels). Each condition was duplicated. The number of significantly differentially expressed (DE) transcripts compared to the control condition for each conditioned medium is indicated in the lower panel. Scale bar: 1 mm. **b** Venn diagram indicating the number of significantly differentially expressed (DE) transcripts compared to the control condition in E8-cSGcm, E8-cDRGcm or E15.5-mSGcm-treated IGR-N91 cell aggregates. **c** Unsupervised clustering and corresponding heatmap presenting mRNA expression for NOR-related (left panel) and MES-related (right panel) gene sets of IGR-N91 cell aggregates treated with control medium, E8-cSGcm, E8-cDRGcm or E15.5-mSGcm. The z score for each transcript is color-coded. **d** GO biological processes significantly represented in DE transcripts common to E8-cSGcm, E8-cDRGcm and E15.5-mSGcm-treated IGR-N91 cell aggregates (n = 308 transcripts; N > 5 hits in each biological process related-gene set; hypergeometric test; p < 0.01). Biological processes are color-coded according to 3 major classes: Cell movement & migration; Response to extracellular cues & signaling; Development & morphogenesis. Sectors thickness represents the proportion of hits detected in each GO biological process related-gene set. **e** Gene Set Enrichment Analysis (GSEA) of a collection of 45 gene signatures related to cell motile behaviors in E8-cSGcm-treated IGR-N91 cell aggregates compared to control. Significantly regulated gene signatures (NES > 1.3; Phenotype-based permutation test[72], nom p < 0.05; FDR q < 0.25) are presented in a graphical form. **f** Unsupervised clustering and corresponding heatmap presenting mRNA expression for OLFM-related gene set of IGR-N91 cell aggregates treated with control medium, E8-cSGcm, E8-cDRGcm or E15.5-mSGcm. The z score for each transcript is color-coded.

blockade, the cumulated volume of all metastatic foci was similar between control and OLFM1 Ab-treated embryos (Fig. 5i). This suggested that interfering with OLFM1-mediated signaling alters NB cell escape from the primary tumor and subsequent migratory and invasive steps, leading to an aborted metastatic process, while not affecting the proliferative state of NB cells.

**Blocking OLFM1 signal alters NB metastatic properties in patient-derived xenografts avian models.** As intra- and inter-tumor heterogeneity is a major hallmark of metastatic NB, we studied whether our findings with NB cell lines were transposable to NB patient samples (Supplementary Table 4). First, a stage 4 NB sample from an adrenal primary tumor (NB#1), was

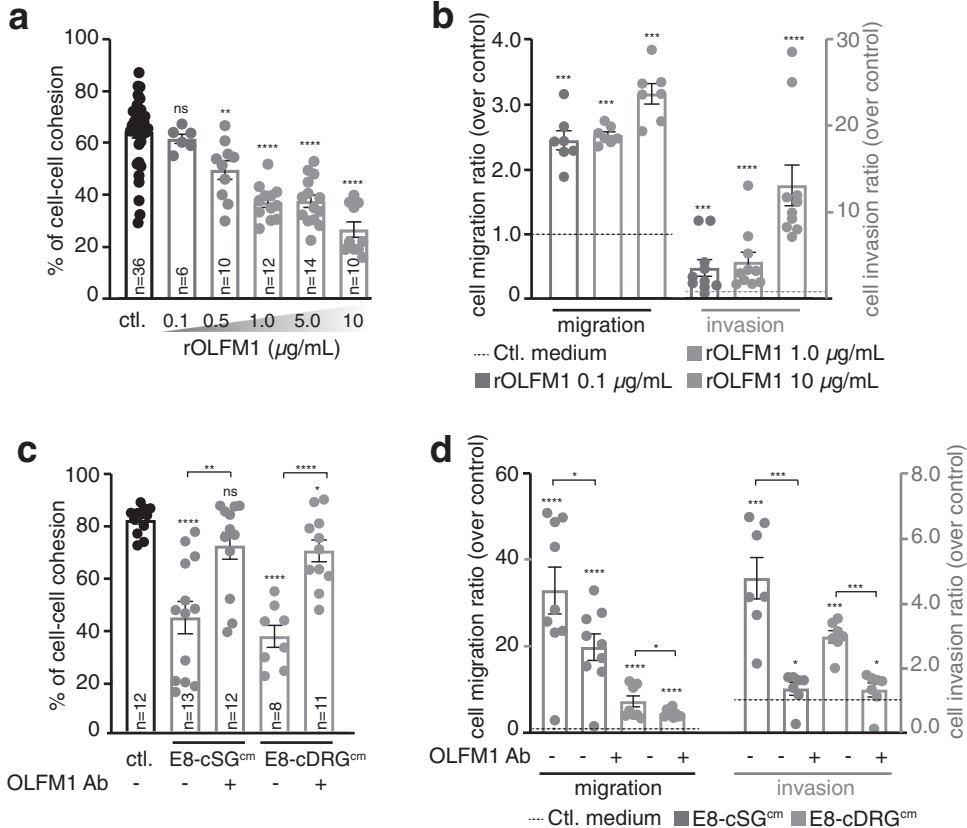

**Fig. 4 Paraspinal-derived OLFM1 boosts NB metastatic properties in vitro. a** Quantification of cell–cell cohesion rate for IGR-N91 cells treated with increasing doses of recombinant OLFM1 (rOLFM1)^γ compared to control medium (ctl.) ($N = 5$ independent experiments; n: number of aggregates analyzed per condition; two-sided Mann–Whitney $U$ test; comparison to ctl.: rOLFM1 0.1 μg/mL: $p = 0.1308$, 0.5 μg/mL: $p = 0.0016$, 1 μg/mL: $p < 0.0001$, 5 μg/mL: $p < 0.0001$, 10 μg/mL: $p < 0.0001$). **b** Quantification of IGR-N91 cells migration and invasion in transwell assays with increasing doses of rOLFM1. Ratios over the control condition are shown ($N = 7$ and $N = 10$ independent experiments for migration and invasion assays; two-sided Mann–Whitney $U$ test; comparison to ctl.: rOLFM1 0.1 μg/mL: $p = 0.0006$, 1 μg/mL: $p = 0.0006$, 10 μg/mL: $p = 0.0006$ for migration; rOLFM1 0.1 μg/mL: $p = 0.0006$, 1 μg/mL: $p < 0.0001$, 10 μg/mL: $p < 0.0001$ for invasion) **c** Quantification of cell-cell cohesion rate of IGR-N91 cells treated with E8-cSG^cm or E8-cDRG^cm supplemented or not with OLFM1 blocking antibody (OLFM1 Ab) ($N = 4$ independent experiments; n: number of aggregates analyzed per condition; two-sided unpaired $t$ test with Welch's correction; comparison to ctl: E8-cSG^cm: $p < 0.0001$, E8-cSG^cm + OLFM1 Ab: $p = 0.0882$, E8-cDRG^cm: $p < 0.0001$, E8-cDRG^cm + OLFM1 Ab: $p = 0.0204$; E8-cSG^cm vs E8-cSG^cm + OLFM1 Ab: $p = 0.0024$; E8-cDRG^cm vs E8-cDRG^cm + OLFM1 Ab: $p < 0.0001$).
**d** Quantification of IGR-N91 cells migration and invasion in transwell assays with E8-cSG^cm or E8-cDRG^cm supplemented or not with OLFM1 Ab. Ratios over the control condition are shown. ($N = 9$ independent experiments for migration, two-sided unpaired $t$ test with Welch's correction; $N = 7$ independent experiments for invasion, two-sided Mann–Whitney $U$ test; comparison to ctl migration/invasion: E8-cSG^cm: $p = 0.0004/p = 0.0006$, E8-cSG^cm + OLFM1 Ab: $p = 0.0003/p = 0.0169$, E8-cDRG^cm: $p = 0.0009/p = 0.0006$, E8-cDRG^cm + OLFM1 Ab: $p < 0.0001/p = 0.0169$; E8-cSG^cm vs E8-cSG^cm + OLFM1 Ab: $p = 0.0511/p = 0.0006$; E8-cDRG^cm vs E8-cDRG^cm + OLFM1 Ab: $p = 0.0486/p = 0.0006$).

dissociated to perform cell aggregation assays (Fig. 6a). Interestingly, exposure of patient cell aggregates to E8-cSG^cm triggered a significant loss of cell–cell cohesion, with clear cell detachment specifically at the edge of the cell aggregate (Fig. 6a, b). This effect was completely blocked upon addition of OLFM1 Ab to E8-cSG^cm. Next, we engrafted two different stage 4 NB patient samples -NB#2 from a primary abdominal mass and NB#3 from a bone marrow aspirate (Supplementary Table 4)- in series of embryos. We treated embryos at E3.5 either with OLFM1 Ab or with the antibody excipient (PBS). E9 embryos were harvested and labelled with an anti-mito antibody specific to human cells for exhaustive detection of NB cells in whole embryos using light sheet microscopy (Fig. 6c, d). For both avian models of Patient Derived Xenografts (PDX) and consistent with our previous findings[40], we observed a wide dissemination of human NB foci in control embryos, with a mean number of 388 metastatic foci for sample NB#2 and 340 for sample NB#3 (Fig. 6e, g). Interestingly, OLFM1 Ab treatment resulted in a significant reduction

of the number of metastatic foci observed in both NB#2 and NB#3 PDX models. In contrast, we found a reduction of the mean distance of secondary foci from the primary tumor site for NB#2 but not for NB#3 (Fig. 6f, h). Such functional differences among NB patients exemplify reported variations and underlying heterogeneity of NBs, that might be driven by etiological or tumoral events specificities. In support of this observation, we analyzed the expression of OLFM-related gene set in previously published single cell RNASeq data[5] and found it was expressed throughout the human fetal adrenal lineages. Interestingly, the above-mentioned heterogeneity was also manifested in the variability of expression of this gene set in each represented physiological cell type, particularly within the neuroblastic sublineages (Supplementary Fig. 6a–c). It was even more pronounced at an early stage of fetal adrenal gland development (8 weeks post-conception) (Supplementary Fig. 6d–f). Moreover, we found differential representation and expression levels of individual genes among the cell identities, providing a basis for the variety of functional

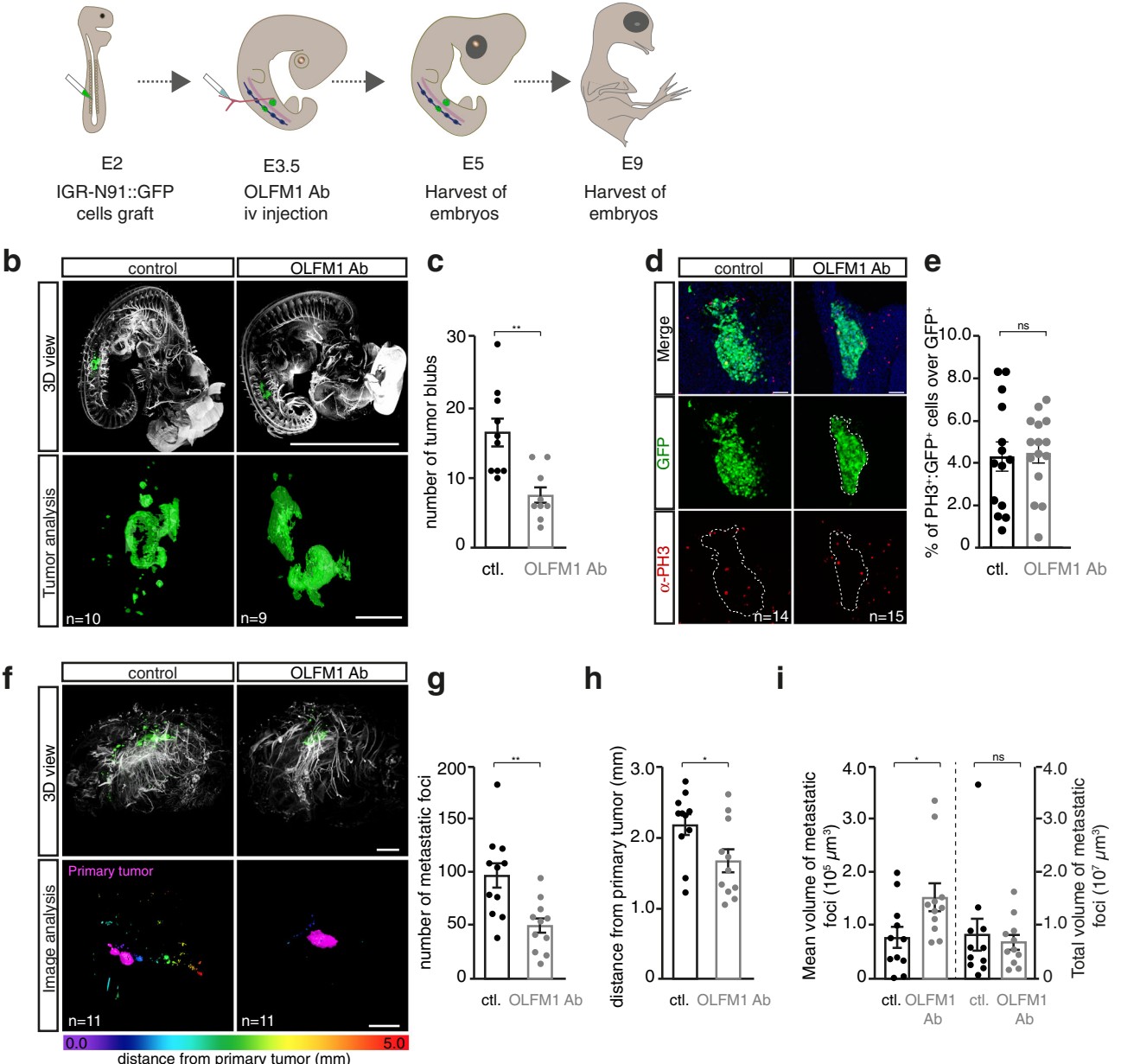

**Fig. 5 Paraspinal-derived OLFM1 boosts NB metastatic properties in vivo. a** Overview of the in vivo grafting procedure. E: Embryonic day. **b** 3D-imaging of E5 chick embryo treated or not with OLFM1 Ab and labelled with α-NF160 (nervous tracts) and α-GFP (IGR-N91 cells) antibodies. Scale bar: 250 μm. **c** Quantification of the mean number of tumor buds detaching from primary tumor masses ($n = 10$ control and $n = 9$ OLFM1 Ab-treated embryos; two-sided unpaired $t$ test; $p = 0.0016$). **d, e** Representative images (**d**) and quantification (**e**) of α-PH3 (mitoses) and α-GFP (IGR-N91 cells) immunofluorescence on cryosections of E5 grafted chick embryos treated or not with OLFM1 Ab. The ratio of PH3$^+$/GFP$^+$ double positive cells over GFP$^+$ cells is quantified ($n = 14$ slices from 5 control embryos and $n = 15$ slices from 5 OLFM1 Ab-treated embryos; two-sided unpaired $t$ test, $p = 0.8381$). Scale bar: 100 μm. **f** 3D-imaging of E9 chick embryo treated or not with OLFM1 Ab and labelled with α-NF160 and α-GFP (IGR-N91 cells) antibodies (upper panels). Lower panels illustrate identification of the primary tumor (in pink) and metastatic foci color-coded according to their distance to the primary tumor. Scale bar: 1 mm. **g–i** Quantification of the mean number of metastatic foci (**g**), mean distance of metastatic foci from the primary tumor (**h**) and volume of metastatic foci (**i**) in $n = 11$ control versus $n = 11$ OLFM1 Ab-treated embryo at E9. In i, mean volume of metastatic foci (left axis) and total volume occupied by metastatic foci (right axis) are presented. Two-sided Mann–Whitney tests were performed. In (**g**), $p = 0.0035$; in (**h**), $p = 0.0288$; in (**i**), $p = 0.0336$ for mean volumes and $p = 0.9487$ for total volumes of metastatic foci. Error bars show SEM. Source data are provided as a Source Data file.

responses to OLFM1. Analysis of another set of scRNASeq data from 16 neuroblastoma human tumors[3] confirmed that the range of expression of the OLFM-related gene set was not only highly heterogeneous among tumor cells (Supplementary Fig. 6g, h) but also between each tumor sample (Supplementary Fig. 6i).

**Blocking OLFM1 impacts on NB dissemination but has no such outcome on sympathetic neurons.** Physiological and pathological contributions of OLFM proteins, and more particularly of OLFM1, remain poorly known. Thus, we assessed whether manipulations of OLFM1 have comparable outcomes on

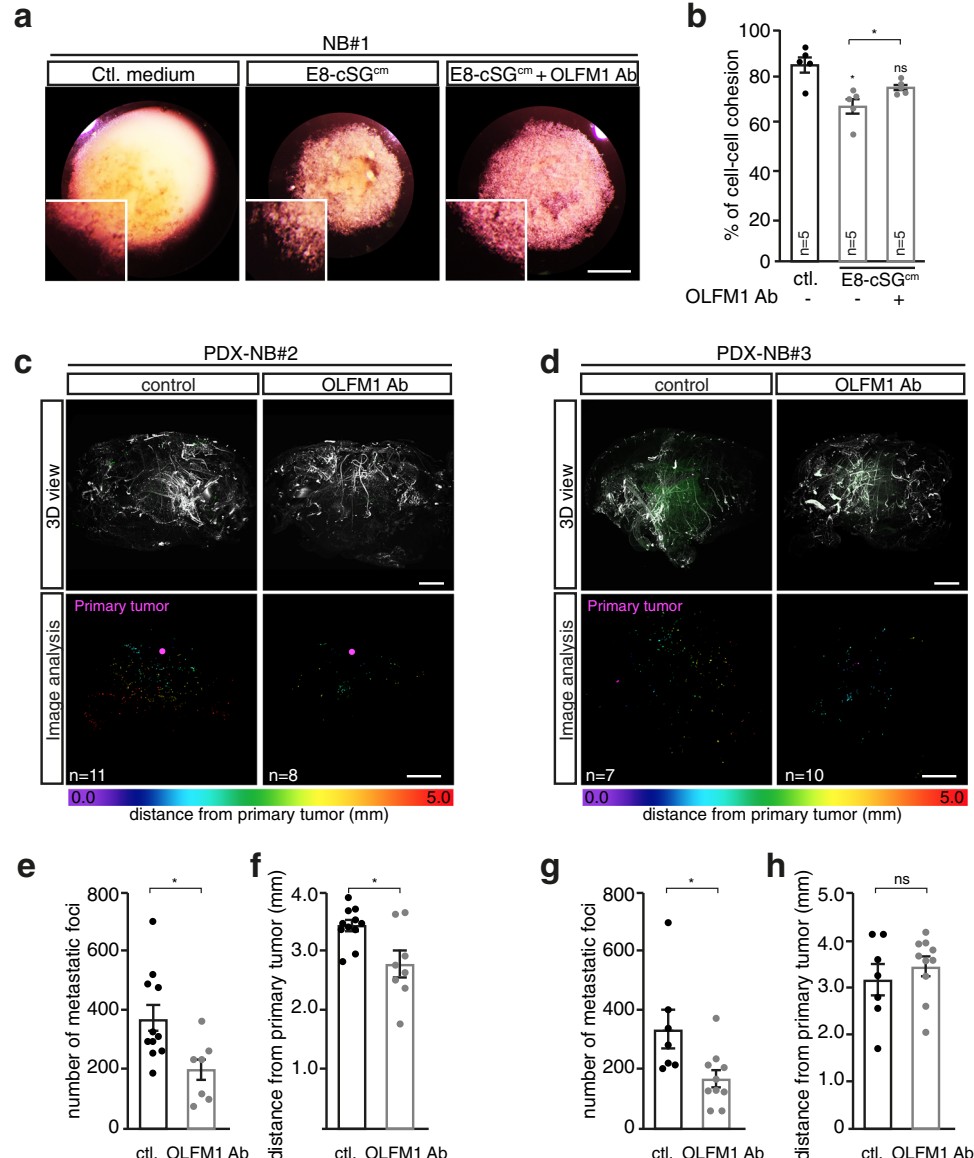

**Fig. 6 Sympathetic-derived OLFM1 triggers NB patient cells escape from the primary tumor. a** Representative pictures of NB#1-patient cell aggregates treated with E8-cSG$^{cm}$ supplemented or not with OLFM1 blocking antibody (OLFM1 Ab). Scale bar: 1 mm. **b** Quantification of cell-cell cohesion rate for NB#1-patient dissociated cells, cultured in hanging drops and treated with E8-cSG$^{cm}$ supplemented or not with OLFM1 Ab ($N = 5$ independent experiments; two-sided Mann–Whitney $U$ test; comparison to ctl: E8-cSG$^{cm}$: $p = 0.0159$, E8-cSG$^{cm}$ + OLFM1 Ab: $p = 0.0952$, E8-cSG$^{cm}$ vs E8-cSG$^{cm}$ + OLFM1 Ab: $p = 0.0317$). **c, d** 3D lightsheet confocal imaging of E9 chick embryos grafted with NB#2 (**c**) or NB#3 (**d**) patient cells and treated or not with OLFM1 Ab. Embryos were labelled with α-NF160 and α-mito (patient cells) antibodies (upper panels). Lower panels illustrate 3D image analysis with identification of the primary tumor sites (in pink) and metastatic foci color-coded according to their distance to the primary tumor site. Scale bar: 1 mm. **e–h** Quantification of the mean number of metastatic foci (**e, g**) and mean distance of metastatic foci from the primary tumor site (**f, h**) in control versus OLFM1 Ab-treated embryo at E9 grafted with NB#2 (**e, f**; $n = 11$ control and $n = 8$ OLFM1 Ab-treated embryos; two-sided unpaired $t$ test; in (**e**): $p = 0.0113$, in (**f**): $p = 0.0241$) or NB#3 (**g, h**; $n = 7$ control and $n = 10$ OLFM1 Ab-treated embryos; two-sided Mann–Whitney $U$ test; in (**g**): $p = 0.0136$, in (**h**): $p = 0.6009$) patient cells. Error bars show SEM. Source data are provided as a Source Data file.

sympathetic neurons and on NB cells. First, we performed aggregation assays with cells dissociated from chick sympathetic chains harvested at E8. Addition of E8-cSG$^{cm}$ supplemented or not with OLFM1 Ab had no impact on the sympathetic cell aggregation rate, in sharp contrast to what we observed with NB cells (Fig. 7a, b). Moreover, mouse recombinant OLFM1 added to either E8 chick or E15.5 mouse sympathetic cell aggregates did not alter their cohesion either (Supplementary Fig. 7a–d).

Second, we assessed whether blocking OLFM1 at trunk NCCs post-migratory stages could affect sympathoadrenal morphogenesis. We analyzed the sympathetic chains of E5 and E9 avian embryos treated with OLFM1 Ab versus excipient as for NB cells graft experiments. On serial cryosections, we performed immunolabeling of HNK1, a marker mainly expressed by migrating and early-post migrating neural crest cells[43,44] and of Phox2b, a key transcription factor involved in the differentiation program of sympathetic neurons[45]. Analysis of the cumulative area covered by HNK1$^+$ immature sympathetic ganglia revealed no difference between experimental conditions (Fig. 7c, d). Similarly, Phox2b expression pattern was identical (Fig. 7c). 3D light sheet confocal imaging of E9 embryos following whole mount immunofluorescence of HNK1 marker also revealed no significant differences in

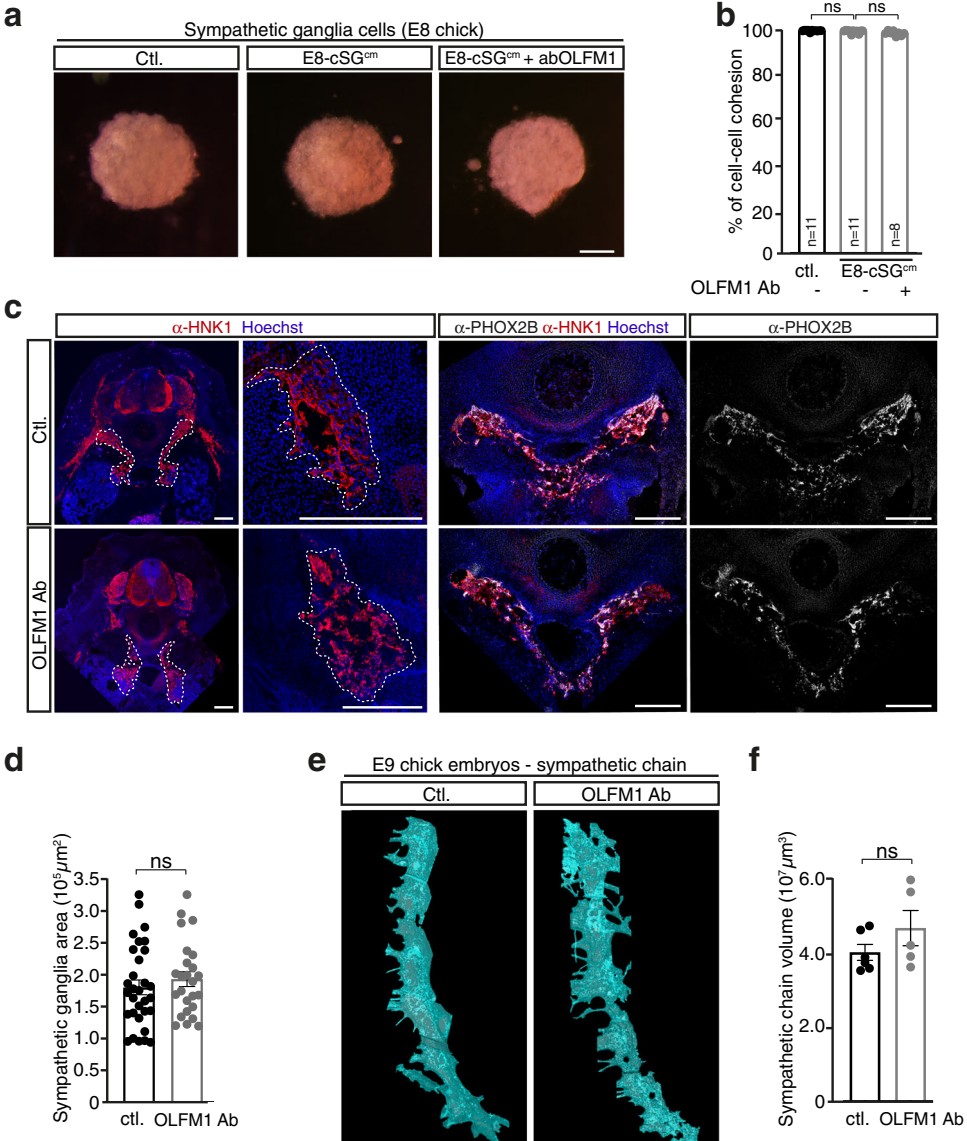

**Fig. 7 Blocking sympathetic-derived OLFM1 doesn't affect the course of sympathetic ganglia cohesion and global morphogenesis. a** Representative pictures of cell aggregates obtained from dissociated E8 chick sympathetic ganglia treated with E8-cSGcm supplemented or not with OLFM1 Ab. Scale bar: 1 mm. **b** Quantification of cell-cell cohesion rate for dissociated cells from E8 sympathetic ganglionic chains, cultured in hanging drops and treated with E8-cSGcm supplemented or not with OLFM1 Ab compared to control medium (ctl.). (N = 3 independent experiments; n: number of aggregates analyzed per condition; two-sided Mann–Whitney U test; ctl vs E8-cSGcm: p = 0.0991, E8-cSGcm vs E8-cSGcm + OLFM1 Ab: p = 0.1682). **c** Representative images of α-HNK1 (neural crest-derived cells) and α-Phox2b immunofluorescence on cryosections performed with E5.5 chick embryos treated or not with OLFM1 Ab at E3.5. Scale bar: 200 μm. **d** Quantification of the mean area covered by sympathetic ganglia labeled with HNK1 immunofluorescent labelling as illustrated in (**c**) (n = 31 sections from 5 ctl. embryos and n = 25 sections from 5 OLFM1 Ab-treated embryos; two-sided unpaired t test, p = 0.4339). **e** Representative illustrations of E9 chick sympathetic chains (6 caudal-most sympathetic ganglia) obtained from 3D reconstruction of lightsheet confocal imaging of HNK1 immunostaining performed on E9 chick embryos treated or not at E3.5 with OLFM1 Ab. Scale bar: 200 μm. **f** Quantification of the mean volume occupied by the 6 caudal most sympathetic ganglia outlined by HNK1 immunostaining in E9 chick embryos treated or not at E3.5 with OLFM1 Ab (**e**) (n = 6 control and n = 5 OLFM1 Ab-treated embryos; two-sided Mann–Whitney U test; p = 0.2468). Error bars show SEM. Source data are provided as a Source Data file.

the total volume occupied by ganglia nor in their morphology (Fig. 7e, f). Thus, transiently blocking OLFM1 at E3.5 in the chick embryo does not alter the global morphogenetic program of sympathetic ganglia while strongly regulating NB cohesive, migratory and invasive properties.

**Metastatic NB share a "primary tumor cell escape" signature with genes related to metastatic neural crest-derived cancers.** Next, we searched whether a restricted gene set extracted from our RNAseq data could feature NB metastatic onset -ie, tumor

cell escape from the primary tumor- in metastatic NB patients. Among the 308 transcripts differentially expressed in NB cells upon DRGcm/SGcm exposure, we extracted a minimal signature of 93 genes that significantly discriminated stage 4 NB from lower stages (1-2-3) in two different published cohorts of human NB (E-GEOD-45547 and GSE85047) (Fig. 8a, b and Supplementary Table 5). Notably, upregulated and downregulated genes of the primary tumor cell escape signature were also significantly regulated between stage 4 NB and stages 1-2-3 in two other independent cohorts (GSE120572 and E-MTAB-8248) (Fig. 8c).

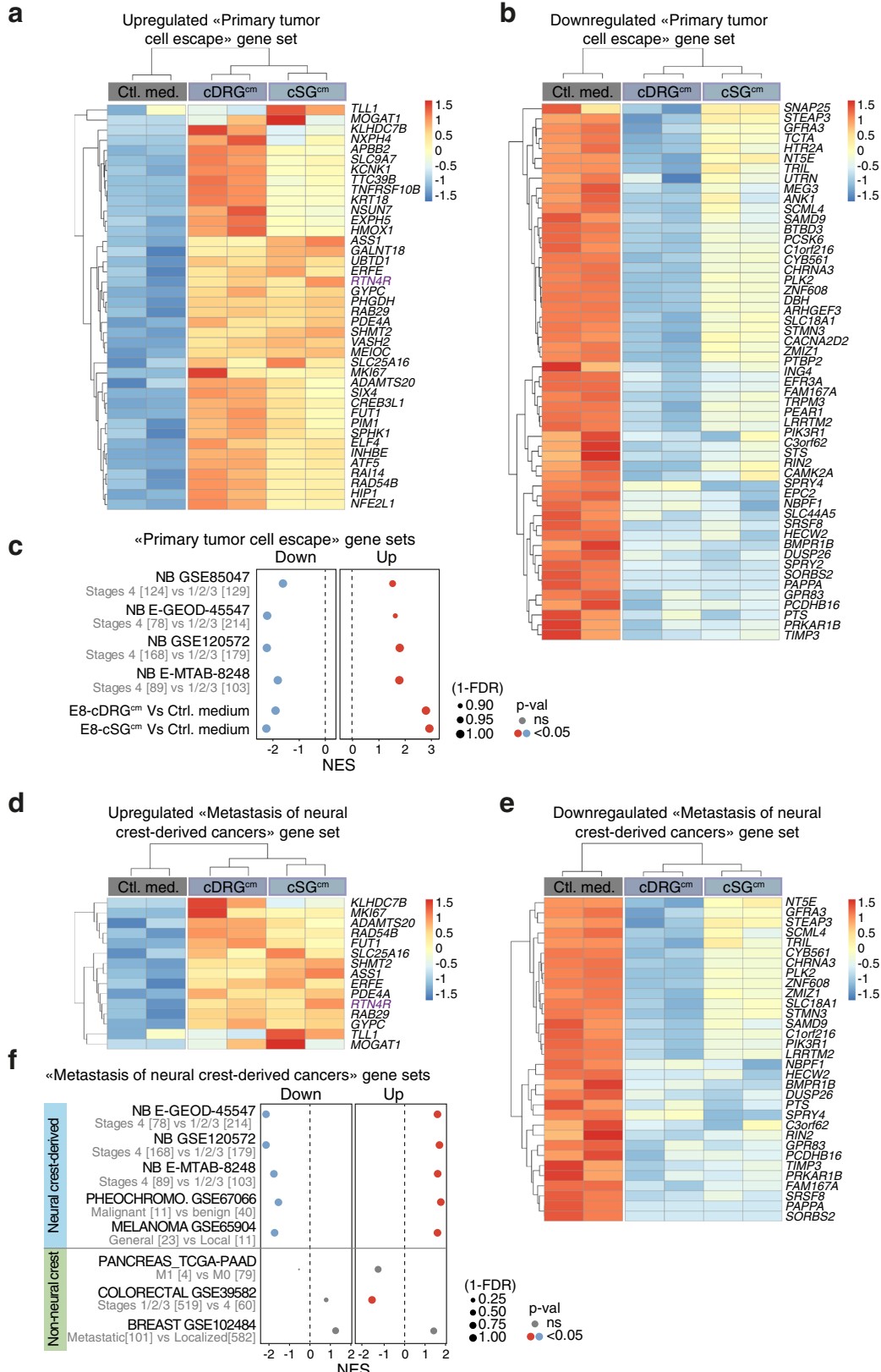

Then, we used two different gene signatures recently shown to feature MES/NOR identities: restricted MES/NOR-related gene sets defined as minimal signatures and exhaustive MES/NOR gene sets combining data from recent publications (Supplementary Data 1). We found that upregulated genes of the primary tumor cell escape signature positively correlated with both restricted and exhaustive MES-related gene sets (Supplementary Fig. 8a, b) in two distinct NB patient cohorts (GSE62564 and GSE45547). Conversely, in the same cohorts, upregulated genes of the primary tumor cell escape signature negatively correlated with NOR-related gene sets (Supplementary Fig. 8c, d and Supplementary Data 1). Interestingly, at the single cell level, upregulated

**Fig. 8 A primary tumor cell escape gene signature outlines the metastatic features of neural crest-related cancers. a, b** Unsupervised clustering and corresponding heatmap using mRNA expression for upregulated (**a**) and downregulated (**b**) "primary tumor cell escape" gene sets in IGR-N91 cell aggregates treated with control medium, E8-cSG$^{cm}$ or E8-cDRG$^{cm}$. The z-score for each transcript is color-coded. Each experimental condition was sequenced in duplicate. **c** Scores obtained from enrichment plots analyzing the behavior of the "primary tumor cell escape" gene sets in neuroblastoma cohorts (GSE85047, E-GEOD-45547, GSE120572, E-MTAB-8248) and in E8-cSG$^{cm}$- and E8-cDRG$^{cm}$-treated IGR-N91 cell aggregates compared to the control condition. For NB, comparison between loco-regional NBs and stage 4 metastatic NBs are performed (Phenotype-based permutation test[72], nom $p < 0.05$; FDR $q < 0.25$). **d, e** Unsupervised clustering and corresponding heatmap using mRNA expression for upregulated (**d**) and downregulated (**e**) "metastasis of neural crest-derived cancer" gene sets in IGR-N91 cell aggregates treated with control medium, E8-cSG$^{cm}$ or E8-cDRG$^{cm}$. The z score for each transcript is color-coded. Each experimental condition was sequenced in duplicate. **f** Scores obtained from enrichment plots analyzing the behavior of "metastasis of neural crest-derived cancer" gene sets in published cohorts of patients having neural crest-related cancers, -ie: neuroblastoma, melanoma or pheochromocytoma or non-neural crest-related cancers -ie: colorectal, breast and pancreatic cancers-. For NB, comparison between loco-regional NBs and stage 4 metastatic NBs are performed in three cohorts (E-GEOD-45547, GSE120572, E-MTAB-8248). For melanoma, local melanomas were compared to general cases (GSE65904). Pheochromocytomas (GSE67066) were confronted according to their benign or malignant phenotype. For colorectal (GSE39582), breast (GSE102484) and pancreatic (TCGA_PAAD) cancers, local versus metastatic cases were confronted (Phenotype-based permutation test[72], nom $p < 0.05$; FDR $q < 0.25$).

genes of the signature were expressed at significantly higher levels in NB cells assigned with a MES identity than in NB cells having a NOR identity (Supplementary Fig. 8e–g) even if the relative proportion of cells assigned with a MES or NOR identity varied according to the restricted/exhaustive gene sets used.

Using the same logic, we thought to identify a restricted gene set that would more generally feature the metastatic onset in neural-crest derived cancers. We extracted from our NB primary tumor cell escape dataset the combination of genes leading to the most significant enrichment score when comparing local and metastatic melanomas (GSE65904 cohort), another emblematic NC-derived cancer[46] (Fig. 8d, e and Supplementary Table 5). Interestingly, these upregulated/downregulated gene signatures were also found significantly regulated in a cohort of pheochromocytoma (GSE67066), another adult NC-derived cancer[46], distinguishing malignant from benign cases (Fig. 8f). We next assessed the signature in patient tumors unrelated to the NC but with known histopathological signs of tumor budding in metastatic forms (1 pancreatic, 1 breast and 1 colorectal cancer cohorts). In these non-NC cancer cohorts, we observed no clear regulation of our gene sets in metastatic versus localized cases (Fig. 8f).

**The OLFM1 receptor RTN4R/NogoR is required for loss of NB cell–cell cohesion and pro-migratory effects of sympathetic cues.** In upregulated genes of both NB primary tumor escape and metastasis of NC-derived cancers signatures, we noticed RTN4R/NogoR, one of the first characterized direct receptor of OLFM1[23] (Fig. 8a, d). Interestingly, in NB patient cohorts, high levels of *RTN4R* but not of two other known genes encoding receptors of OLFM1, *APP* and *GRIA2/AMPAR* were found associated with a lower overall survival (Fig. 9a–c and Supplementary Fig. 9a–c). We then assessed the contribution of these three receptors to NB cell responses to sympathetic cues, using a siRNA-based approach (Supplementary Fig. 9d). In cell aggregation assays, we found that *RTN4R* but not *APP* and *GRIA2* downregulation abolished E8-cSG$^{cm}$-mediated loss of cell–cell cohesion (Fig. 9d, e). Similar results were obtained with two different *RTN4R* siRNAs (siRNA RTN4Ra and siRNA RTN4Rb) in both IGR-N91 and SH-SY5Y cell lines (Supplementary Fig. 9e–h). *RTN4R* downregulation also abolished recombinant OLFM1-mediated loss of cell–cell cohesion, and this effect was rescued by co-transfection with a vector encoding for human RTN4R (Fig. 9f, g and Supplementary Fig. 9i). Similarly, *RTN4R* downregulation abolished the pro-migratory effect of E8-cSG$^{cm}$ on NB cells (Fig. 9h and Supplementary Fig. 9j). Moreover, a polyclonal RTN4R antibody reported to block RTN4R extracellular ligand binding sites[47,48] mimicked the action of OLFM1 antibody in cell aggregation

assays, significantly decreasing E8-cSG$^{cm}$-mediated loss of cell-cell cohesion (Fig. 9i, j and Supplementary Fig. 9k, l). Finally, we analyzed the expression of *RTN4R* in single cell dataset of NB tumors. We found heterogeneity, with only a subset of cells displaying high *RTN4R* levels, which suggests that in tumors, not all NB cells might be sensitive to OLFM1 pro-metastatic effects (Supplementary Fig. 9m).

## Discussion

Major roles of the microenvironment in the tumorigenesis process are acknowledged, which considerably complexifies the ancient view of cancer cells as "lonely", unruly invaders[20,49]. Here, we provide the evidence that physiological signals expressed during embryonic peripheral ganglia organogenesis, including OLFM1, are exploited by NB cells to escape from the primary tumor site and engage in metastasis.

First, OLFM1 was present in our proteomic screen of the peripheral ganglia secretome. Second, NB cells reacted to exogenous recombinant OLFM1 exposure by downregulating the contacts with their neighbors in the tumor mass and switching towards a migratory state, acquiring mesenchymal features. Third, function-blocking OLFM1 antibody abolished the anti-cohesion properties of peripheral ganglia conditioned media. Fourth, in vivo, anti-OLFM1 antibody injection efficiently interfered with NB metastatic process. Finally, these functional properties were correlated with a set of genes functionally related to OLFMs, co-regulated in NB cells exposed to embryonic signals.

It remains unclear whether NB cell detachment primed by OLFM1 in the primary tumor influences subsequent steps of NB metastatic progression, particularly NB cell migration properties and dissemination paths. Our in vivo analysis indicates that some NB cells disseminate at lower distance from the primary tumor upon OLFM1 blockade. On the one hand, this could reflect a delay in the detachment and escape from the primary tumor, with cells that succeeded to evade having migration properties similar to those of the control. On the other hand, inhibition of OLFM1 in the primary tumor could imprint a sustained difference on NB cells, conferring them lower migration capabilities. Whatever the case, the targeting of OLFM1 achieved in our studies resulted in significant beneficial consequences on the metastatic disease. Although metastatic stages are already manifested at the diagnosis[50], interfering with OLFM1 could slow-down the progression of disseminating NB cells. Notably, we observed different responses of NB patient samples to OLFM1 blocking antibody in vivo, which could take their root in the heterogeneous expression of OLFM partners in the NB lineage of origin.

Deciphering the diversity of OLFM1 functions and their specific downstream molecular mechanisms in NB will represent a

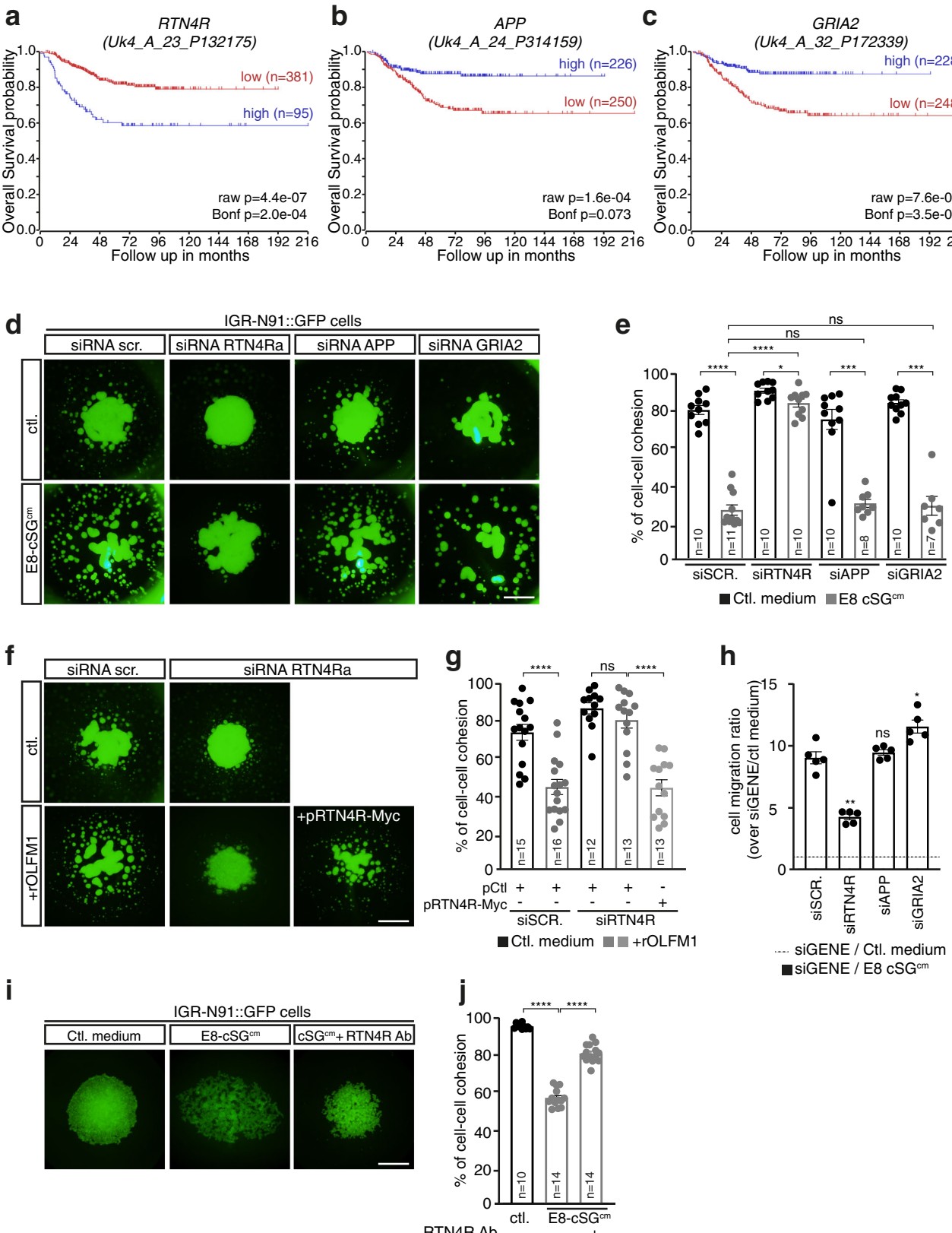

promising approach towards the design of optimal therapeutic tools targeting this pathway. As a first step, we identified RTN4R as a mandatory receptor for mediating the effects of microenvironmental sympathetic cues in NB cells. While interfering with APP and AMPA receptors had no specific outcome, decreasing RTN4R levels or inhibiting its activity with a function-blocking antibody strongly altered sympathetic cues/OLFM1-induced decohesion and migration potential of NB cells. Remarkably in patients' cohorts, only *RTN4R* levels significantly distinguished metastatic NBs from other stages.

OLFM1 is highly enriched in the developing nervous system, particularly in post-delamination NCC and their tissue derivatives[21]

**Fig. 9 The OLFM1 receptor RTN4R is functionally involved in NB loss of cell–cell cohesion and migratory response to sympathetic cues.**
**a–c** Kaplan–Meier analysis of overall survival probability according to *RTN4R* (**a**), *APP* (**b**) and *GRIA2* (**c**) expression levels in Shi and Fischer's published cohort (GEO: GSE62564; http://r2.amc.nl; n = 498 samples). Raw and Bonferroni corrected p values are indicated on the graphs. **d, e** Representative pictures (**d**) and quantification of cell–cell cohesion rate (**e**) of IGR-N91 cell aggregates transfected with a control (siSCR) or a RTN4R siRNA (siRTN4Ra) or an APP siRNA (siAPP) or a GRIA2 siRNA (siGRIA2) and treated with E8-cSG^cm (N = 5 independent experiments; n: number of aggregates analyzed per condition; two-sided Mann–Whitney U test; ctl vs E8-cSG^cm: siScr: p < 0.0001, siRTN4Ra: p = 0.0272, siAPP: p = 0.0003, siGRIA2: p = 0.0001; comparison to E8-cSG^cm/siScr: E8-cSG^cm/siRTN4Ra: p < 0.0001, E8-cSG^cm/siAPP: p = 0.1343, E8-cSG^cm/siAPP: p = 0.8601). **f, g** Representative pictures (**f**) and quantification of cell–cell cohesion rate (**g**) of IGR-N91 cell aggregates transfected with a control (siSCR) or a RTN4R siRNAs (siRTN4Ra, siRTN4Rb) together or not with a vector encoding for human RTN4R (pRTN4R-Myc), and treated with 10 μg/mL rOLFM1 (N = 3 independent experiments; n: number of aggregates analyzed per condition;; two-sided Mann–Whitney U test; ctl vs rOLFM1: siScr: p < 0.0001, siRTN4Ra: p = 0.2455; rOLFM1/siRTN4Ra pCtl vs pRTN4R-Myc: p < 0.0001). **h** Quantification of IGR-N91 cells migration properties in transwell assays using E8-cSG^cm in the lower part of the device. Cells were transfected either with a control (siSCR) or a RTN4R siRNA (siRTN4Ra) or an APP siRNA (siAPP) or a GRIA2 siRNA (siGRIA2). Ratios over the number of migrating cells in the control condition (medium without any cultured tissue in the lower part) are shown (N = 5 independent experiments, two-sided Mann–Whitney U test; comparison to siScr: siRTN4Ra: p = 0.0079, siAPP: p = 0.4206, siGRIA2: p = 0.0159). **i, j** Representative pictures (**i**) and quantification of cell-cell cohesion rate (**j**) of IGR-N91 cell aggregates treated with E8-cSG^cm supplemented or not with 50 μg/mL RTN4R antibody (RTN4R Ab) (N = 3 independent experiments; n: number of aggregates analyzed per condition; using two-sided unpaired t test with Welch's correction; ctl vs E8-cSG^cm: p < 0.0001, E8-cSG^cm vs E8-cSG^cm + RTN4R Ab: p < 0.0001). Error bars show SEM. Scale bar: 1 mm. Source data are provided as a Source Data file.

and consistently, was detected both in our analysis of sympathetic ganglia secretome and by immunolabeling embryonic sections. Moreover, our findings that OLFM1 promotes the expression of NB cells mesenchymal potential echoes with its reported activity during heart valve morphogenesis in promoting the generation of mesenchymal cells from heart endothelium and their related invasive properties[24]. This raises the question of whether NB cell behaviors induced by OLFM1 are reminiscent of those of their related physiological cells. Earlier work reported that OLFM1 overexpression in the chick neural tube resulted in excess NCC emigration and in prolongation of the timing of NCC production[21]. Some of these phenotypes could result from loss of cell-cell cohesion and acquisition of pro-migratory abilities, thus resembling behaviors observed for their NB pathological counterparts.

Interestingly, however, OLFM1 manipulations altered neither the cohesion state of sympathetic cells in our ex vivo assays nor the morphology of the nascent sympathetic territory and the resulting ganglia in vivo. Thus, physiological OLFM1 activities – that could potentially involve RTN4R- might regulate other processes of the sympathetic gangliogenesis that remain to be assessed. However, these findings support that NB cells engaging in metastatic progression reactivate or retain behaviors reminiscent of their initial cell of origin. The functions ensured by OLFM1 during peripheral nerve ganglia differentiation remain to be understood. Meanwhile, precisely decoding OFLM1 roles in this physiological context will be crucial to discriminate key druggable NB features related to OLFM1 pathway(s).

We found that media conditioned with both SG and DRG had similar outcomes on NB cell migratory properties and gene programs underlying NB switch towards a metastatic phenotype. While both types of ganglia are part of the paraspinal environment, they arise from different lineages of the trunk NC. Moreover, NB malignant transformation is thought to occur in the SG but not DRG lineage, with SG but not DRG hosting primary tumors. Given that these ganglia develop at close proximity and are directly connected via the white and grey ramus communicans, the DRG OLFM1 source could act complementary and subsequently to that of the SG, sustaining NB dissemination via adjacent paths. Interestingly, it was described that key signaling partners produced by DRG such as periostin instruct SCPs to allow their migration on sensitive and motor nerves[51].

Exposing NB cells to signals from embryonic tissues homolog to those in which they emerged in the patients allowed us to examine the plasticity of NB cell phenomes. Recent work revealed that NB cells inherently display two different noradrenergic

(NOR) and mesenchymal (MES) plastic identities, that can interconvert to each other[3]. These co-existing identities are thought to sustain distinct functional states, the MES identity being suspected to mediate NB aggressive behaviors, driving metastatic progression and relapse. Our analyses reveal that the NOR and MES signatures of NB cells can be regulated at the contact of embryonic sympathetic signals with transcriptional and protein down-regulation of key genes of the NOR identity and acquisition of MES features. Interestingly, these changes are consistent with our observed modifications of NB cells behaviors, manifested in their loss of cell-cell cohesion and motility states. This supports that NB secondary dissemination is promoted by morphogenesis signals triggering gene programs associated with their MES identity. Accordingly, we characterized a list of gene regulations associated with NB metastatic status and found that these transcriptomic signatures positively correlated with the expression of MES genes in NB patients' cohorts.

Lastly, our characterization of a signature reflecting early steps of NB metastatic disease also revealed that it contains a minimal set of gene regulations shared between cancers with a NC origin while being inconsistent for non-NC-derived cancers. Multiple theoretical parallels between NCCs properties and metastatic features have been documented[52,53]. In contrast, much less is known on whether NC-derived cancers share pro-metastatic strategies. Our data suggest it might be the case as we found that whatever the NC-derived tissue they emerge from, metastatic malignant cells set common gene regulations upon the metastatic process. Thus, while providing an original view of the NB metastatic cascade, these findings may also lay the bases of specific aspects of NC-derived cancers metastasis and open perspectives in the understanding and therapeutic care of this set of malignancies.

## Methods

**Chick embryos.** Naked Neck strain embryonated eggs were obtained from a local supplier (Elevage avicole du Grand Buisson, Saint Maurice sur Dargoire, France). Laying hen's sanitary status was regularly checked by the supplier according to French laws. Experiments with chick embryos were performed within the 10 first days of gestation, stages that do not require approved protocol by ethics committee, according to the revised European ethics legislation (2013). Eggs were housed in an 18 °C-incubator until use. They were then incubated at 38.5 °C in a humidified incubator until the desired developmental stage, i.e., HH14 for the graft step (54 h of incubation), E3.5 stages (3.5 days of incubation) for the intravenous injection and HH35 for 7 days post-graft analyses.

**Cell lines.** Human stage 4 neuroblastoma IGR-N91 cells were a kind gift of Dr. J. Bénard (Gustave Roussy Institute, Paris, France). SH-SY5Y were obtained from ATCC (ATCC® CRL-2266™). IGR-N91 and SH-SY5Y cell lines stably expressing

GFP (IGR-N91::GFP and SH-SY5Y::GFP) were previously generated and described in[40]. IGR-N91 and SH-SY5Y NB cell lines were cultured in Dulbecco's Modified Eagle Medium (DMEM) GlutaMAX™ (Life Technologies), and SHEP cells were cultured in RPMI 1640 GlutaMAX™ (Life Technologies). Media were each supplemented with 10% Fetal Bovine Serum (FBS), 25 U/mL Penicillin Streptomycin (Sigma), 2.5 μg/mL Amphotericin B (Sigma). Cell lines were maintained in sterile conditions in a 37 °C, 5% $CO_2$-incubator. Cell lines have not been re-authenticated for the present paper.

**Human NB patient samples**. Neuroblastoma patient samples were obtained from patients treated in Centre Léon Bérard and Hopital Femme-Mère-Enfant (Lyon, France). Parents provided written consent for tumour banking and future research use according to national regulations. Studies were authorized by the ethics committees "Comité de Protection des Personnes Sud-Est IV" (L07-95 and L12-171), as well as "Comité de Protection des Personnes de Paris IIe de France I" (ref 08–11728). The samples used for graft experiments and cell aggregation assays in the present paper are listed in Supplementary Table 4.

**Preparation of NB patient samples**. Samples obtained from primary tumor resection (NB#1 and NB#2) were dissociated. Briefly, tissues were washed with $Ca^{2+}$-, $Mg^{2+}$-fisher-free PBS (Life Technologies), crushed with a scalpel in F12 medium (ThermoFisher Scientific) and then dissociated with 156 Units/ml type IV Collagenase (Sigma–Aldrich), 25 mM HEPES (Life technologies), 3% fetal calf serum (FCS, Sigma) and 20 Units/ml Dnase I (Sigma) for 30 min at 37 C. A 5 min more incubation with 4.9 mg/ml trypsin (Sigma) at 37 °C was performed. Non-dissociated tissue was removed by filtration through a 40 mm nylon Cell Strainer (BD Falcon).

Bone marrow sample (NB#3) was harvested under general anesthesia at diagnosis, taken from posterior and anterior iliac crests and collected on EDTA tube. Mononuclear marrow cells were separated by density gradient centrifugation (Pancoll, PAN-Biotech). The percentage of malignant cells in the samples was evaluated by a two-colour fluorochrome staining as described in[54]. The mononuclear marrow cells showing tumoral invasion were frozen in RPMI containing 20% PBS and 10% DMSO. Cell suspensions obtained from dissociated tumors or bone marrow aspirates were labeled with an 8 μM CFSE solution (Life Technologies).

**Plasmids, siRNAs, cell transfection**. Control siRNA (siRNA scr) (siRNA Universal Negative Control #1 SIC001, Sigma-Aldrich), human RTN4Ra (NM_023004; SASI_Hs01_00129206, Sigma-Aldrich), RTN4Rb (NM_023004, SASI_Hs01_00229277, Sigma-Aldrich), APP (NM_000484, SASI_Hs01_00185800, Sigma-Aldrich), GRIA2 (NM_001083620, SASI_Hs01_00035395, Sigma-Aldrich). pRTN4R-Myc, human tagged ORF clone, was purchased from Origene (NM_023004; #RC204073). siRNAs were used at a concentration of 50 nM.

For siRNA and plasmids transfection, cells were transfected with JetPrime according to the manufacturer's guidelines (PolyPlus).

**Production of conditioned medium**. The embryos were dissected out at the desired developmental stages to retrieve Sympathetic Ganglia (SG) or dorsal root ganglia (DRG). Tissues were incubated for 30 min at 37 °C in DMEM medium. Dulbecco's Modified Eagle Medium (DMEM) GlutaMAX™ (Life Technologies), supplemented with 10% Fetal Bovine Serum (FBS), 25 U/mL Penicillin Streptomycin (Sigma), 2.5 μg/mL Amphotericin B (Sigma), was added to the sympathetic ganglia or DRG embedded in 0.5% agarose medium (DMEM 10% FBS, 0.5% agarose low gelling (Sigma, #A9414)). 500 μL of complete culture medium was added for about 15-dissected-embryos-equivalent tissues. 48 h later, the conditioned media were retrieved to be stored at −80 °C. Alternatively, E15.5 mice embryos were dissected out following the exact same protocol.

**Cell aggregation assays**

*NB cell lines aggregation assay*. Fluorescent NB cells were resuspended at a concentration of $5.10^6$ cells/mL and cultured in 25 μL hanging drops on the inside of the top of a culture dish. Cells were resuspended either in control medium or in conditioned media (SG$^{cm}$, DRG$^{cm}$), or in other conditions such as complete medium supplemented with different concentrations of the recombinant protein OLFM1 (R&D Systems, #4636-NL) or SG$^{cm}$ and DRG$^{cm}$ supplemented with the OLFM1 blocking antibody (R&D Systems, #AF4636), the normal sheep IgG control antibody (R&D Systems, ##5-001-A, 10 μg/mL) or the anti-hNogoR antibody (R&D Systems, #AF1208, 50 μg/mL).

Hanging drops were imaged 21 h later under a fluorescence stereomicroscope with QImaging camera MicroPublisher 5.0 (GTVision). The aggregation rate was quantified with ImageJ software. Briefly, fluorescent versus non-fluorescent (black) areas were measured using the « analyze particles » pluggin for each time point. The homogeneity of cell suspension between conditions at T0 was checked.

*Sympathetic ganglia cells aggregation assay*. Sympathetic ganglia were dissected out from either E8 chicken embryos or E15.5 mice embryos and were dissociated in 37 °C-HBSS medium supplemented with 0,13% of trypsine and incubated for

10 min at 37 °C. 0,01% of DNase I was added and after a 10 min-incubation, DMEM medium complemented with 10% FBS was added. After a minute of 250 g-centrifugation, the supernatant was removed and cells manually dissociated by pipetting up and down. Sympathetic ganglia cells were resuspended at a concentration of $10.10^6$ cells/mL with or without recombinant OLFM1 (R&D Systems, #4636-NL) at 10 μg/ml and cultured in 25 μL hanging drops on the inside of the top of a culture dish.

*Patients samples aggregation assay*. Patient samples were prepared as for sympathetic ganglia. The aggregation rate was quantified with ImageJ software focusing on the bordure of the aggregates that was automatically selected, using the "enlarge" pluggin to determine the internal border of the peripheral region. The outer border of this region was drawn by hand. Then, differential areas were measured using the « analyze particles » pluggin.

*Migration assays*. 600 μL of "candidate" medium were put per well into a 24 well-plate before adding the upper chamber of the transwell culture dish (8 μm pore size; BD Falcon, NJ) in which fluorescent NB cells were plated ($5.10^4$ cells/ml). The cells were then incubated for 60 h in a 37 °C, 5% $CO_2$-incubator. Cells retained on the upper face of the membrane were scrubbed using cotton swabs. The transwell culture dishes were then fixed with a 30 min-4% PFA-bath, before washing with 3 successive PBS-baths and mounting in Mowiol. Migrating cells were counted using a confocal microscope (Olympus, FV1000, X81).

*Invasion assays*. The same protocol was applied using transwell culture dish layered with Matrigel (Corning® BioCoat™ Matrigel® Invasion Chambers with 8.0 μm PET Membrane, #354480) and 750 μL of medium in the lower chambers.

*Immunofluorescence on slices*. Chick embryos of interest were harvested and fixed in 4% Paraformaldehyde (PFA). Embryos were embedded in 7.5% gelatin- 15% sucrose in PBS to perform 20 μm transverse cryosections. Permeabilization and saturation of sections was performed in PBS-BSA 3%-Triton 0.5%. The following primary antibodies were applied to sections: anti-HNK1 mouse IgM (1/50, 3H5, DSHB), anti-Phospho Histone 3 (PH3) rat IgG (1/500, Millipore/Sigma, #MABE76), and anti-OLFM1 (rabbit 1/200; Sigma #HPA057444).

The antibody for chicken Phox2b (rabbit, 1/500), was raised against the C-terminal peptide [(Y)PGGTKGSLVKSGMF] of Phox2b from Pseudopodoces humilis (Tibetan ground-tit)[55], originally the only avian *Phox2b* gene whose complete open reading frame was correctly predicted from the genome, as judged from the alignment with vertebrate orthologues — and which later turned out identical to the C-terminal peptide encoded by the corrected Phox2b open reading frame for *Gallus gallus* (Ensembl release 86, Gallus_gallus-5).

Alexa 555 anti-mouse IgM (1/500, A21426, Life Technologies), FP547H anti-rat (1/500, Interchim, #FP-SB61110) and Alexa 488 anti-goat IgG (1/500, molecule probes #A11008) were used as secondary antibodies. Nuclei were stained with Hoechst (H21486, Invitrogen).

Slices were imaged with a confocal microscope (Olympus, FV1000, X81) using either a 10X objective for whole slice imaging or a 40X objective to focus on PH3, PHOX2B and OLFM1 immunolabeling. PH3$^+$ mitotic cells and GFP cells were counted with ImageJ software.

*Mass spectrometry analysis*. Conditioned media were produced as described above with the exception of the media to fit with mass spectrometry constraints: Ham's F12 medium (Gibco) with N2 supplement (ThermoFisher Scientific, 17502048). The following conditions were analyzed: E6-cSG$^{CM}$ and cDRG$^{CM}$, E8-cSG$^{CM}$ and DRG$^{CM}$, E10-cSG$^{CM}$ and cDRG$^{CM}$, and E15.5-mSG$^{CM}$ and mDRG$^{CM}$. Using an Amicon Ultra-0.5 mL, PMNL 10kD (Merck Millipore, #UFC501096), successive steps of concentrations of these conditioned media were performed. The mass spectrometry analysis was done by the EDyP lab (Biosciences and Biotechnology Institute of Grenoble, France). Proteins were solubilized in Laemmli buffer and heated for 10 min at 95 °C. They were then separated by SDS-PAGE (4–12% NuPAGE, Life Technologies), stained with Coomassie blue R-250 (Bio-Rad) before in-gel digestion using modified trypsin (Promega, sequencing grade) as previously described[56]. The very intense band containing Transferrin was prepared separately from the rest of the sample. Resulting peptides were analyzed by online nanoliquid chromatography coupled to tandem MS (UltiMate 3000 and LTQ-Orbitrap Velos Pro, Thermo Scientific). Peptides were sampled on a 300 μm x 5 mm precolumn (PepMap C18, Thermo Scientific) and separated on a 75 μm x 250 mm column (Reprosil-Pur 120 C18-AQ, 1,9 μm, Dr. Maisch) using a 50-min gradient for the Transferrin bands and a 140-min gradient for the rest of the samples. MS and MS/MS data were acquired using Xcalibur (Thermo Scientific). Peptides and proteins were identified using Mascot (version 2.6) through concomitant searches against Uniprot database (*Gallus gallus* and *Homo sapiens* taxonomies, November 2011 version), classical contaminant database (homemade) and the corresponding reversed databases. The Proline software (v2.1.2, PMID: 32096818) was used to filter the results: conservation of rank 1 peptides, peptide score ≥ 25, peptide length ≥ 7 amino acids, peptide-spectrum-match identification false discovery rate <1% as calculated on scores by employing the reverse database strategy, and minimum of 1 specific peptide per identified protein group. Proline was then used to perform a compilation, grouping and spectral counting-based comparison

of the protein groups identified in the different samples. Proteins from the contaminant database and additional keratins were discarded from the final list of identified proteins.

The mass spectrometry proteomics data have been deposited to the ProteomeXchange Consortium via the PRIDE[57] partner repository with the dataset identifier PXD027499.

*Bulk RNAseq analysis.* IGR-N91 cells were cultured in hanging drops with control, E8-cSG<sup>cm</sup>, E8-cDRG<sup>cm</sup> or E15.5-mSG<sup>cm</sup> as for typical aggregation assays. Aggregates of about 2.7.106 cells were then retrieved 20 h later before storing to −80 °C. Duplicates of each condition were sent to ProfileXpert core facility (Lyon, France). Total RNA was extracted with RNeasy PLUS mini kit (Qiagen) and the quality was checked with a Bioanalyzer 2100 (Agilent, RIN > 8.0). RNAseq libraries were prepared with 50 ng of total RNA with SMARTer® Stranded Total RNA-Seq Kit v2 - Pico Input Mammalian (Takara, Clontech). Samples were sequenced in whole RNAseq using the NextSeq500 illumina Platform (75 bp single read). After demultiplexing (Bcl2fastq v2.17.1.14), trimming and quality check (CutAdapt v1.9.1) reads were mapped using TopHat v.2.1.0 against the human Genome build (hg19). Quantification, enrichment set analysis and differential gene expression analysis were performed in the laboratory. Quantification was done using HTSeq-count software (0.11.3). The enrichment set analysis was carried out using the GSEA software (https://www.gsea-msigdb.org/gsea/index.jsp). The differential analysis was performed with the DESeq2 tool (3.11) with median of ratios normalization that allows comparison between samples. The differential analysis focuses on the comparisons of NB cells in conditioned media (E8-cSG<sup>cm</sup>, E8-cDRG<sup>cm</sup>, E15.5-mSG<sup>cm</sup>) against the control (CTRL). A gene is considered to be upregulated or downregulated when its level of expression varies with a fold change greater than or equal to 1.5 between the two conditions and its adjusted $p < 0.05$. Analysis of main functions concerned by gene expression change was performed with ToppFun [https://toppgene.cchmc.org/enrichment.jsp] and ConsensusPathDB-human from Max Planck Institute of Molecular Genetics [http://cpdb.molgen.mpg.de/].

Raw and analyzed bulk RNASeq data generated in the present article have been deposited in the Gene Expression Omnibus (GEO) under accession number GSE169280.

*RNA isolation and quantitative real-time PCR (qRT-PCR).* For qRT-PCR analysis, total RNA was extracted from cells using the Nucleospin RNAII kit (Macherey-Nagel). One μg of total RNA was reverse-transcribed using the iScript cDNA Synthesis Kit (BioRad). qRT-PCR was performed using the LightCycler480 SYBR-Green I Master1 kit (Roche Life Science) and the CFX Connect Real-Time PCR Detection System (BioRad). The following list of primers was used in the study:

HPRT: for TGACACTGGCAAAACAATGCA/ Rev: GGTCCTTTTCACCAGCAAGCT; GATA3: PrimerPCR SYBR Green Assay: GATA3, Human; qHsaCID0017793

PHOX2B: PrimerPCR SYBR Green Assay: PHOX2B, Human; qHsaCED0043265

HAND1: PrimerPCR SYBR Green Assay: HAND1, Human; qHsaCED0043474 HAND2: PrimerPCR SYBR Green Assay: HAND2, Human; qHsaCID0010704 JUN: PrimerPCR SYBR Green Assay: JUN, Human; qHsaCED0018770 MAML2: PrimerPCR SYBR Green Assay: MAML2, Human; qHsaCID0016493 RUNX1: PrimerPCR SYBR Green Assay: RUNX1, Human; qHsaCID0037818

*Proteins isolation for western blot.* Whole cells extract was isolated using RIPA buffer (NaCl 150 mM - Tris HCL pH7.35 50 mM - DOC 1% - NP40 1%) supplemented with protease inhibitor (Ref #04693116001, Roche). Concentration of isolated proteins was determined using Bradford assay (500-0006, Bio-Rad).

Western Blot analysis were performed using the following primary antibodies: anti-GATA3 (1/1000; Ozyme #5852), anti-PHOX2B (1/1000, Ozyme #PA-5115754), anti-c-JUN (1/1000, Cell Signaling #9165), anti-MAML2 (1/500, Sigma–Aldrich clone 4A1 # WH0084441M3), anti-RUNX1 (1/500, Santa Cruz SC-365644), anti-GAPDH (1/500, Sigma–Aldrich #G9545). Anti-mouse IgG HRP (1/10000, Sigma #A4416), anti-rabbit IgG HRP (1/10000, Sigma–Aldrich #A9169) and anti-goat IgG HRP (1/10000, Sigma #A5420) were used as secondary antibodies. Densitometric analyses of western blots were performed using Image J software.

*NB cell lines and patient biopsies graft in ovo.* Stage HH14 chick embryos were grafted with approximately 2500 fluorescent NB cells at the neural crest level, i.e., in the region between the dorsal neural tube and the epidermis, between somite 18 and presumptive somite 24. NB cells were implanted with a glass capillary connected to a pneumatic PicoPump (PV820, World Precision Instruments) under a fluorescence stereomicroscope. Eggs were closed with solvent-free tape and placed back in the humidified incubator until the desired stage.

36 h hours post-graft, embryos were injected intravenously via the chorioallantoic membrane with physiological serum solution (NaCl 0,9% filtrated to ensure sterility), or with 4 mg/kg of OLFM1 blocking antibody from R&D Systems (#AF4636, Minneapolis, MN).

*Tissue clearing, whole-mount immunofluorescence and selective plane illumination microscopy (SPIM) imaging.* PFA-fixed E5 embryos or E9 embryos were dehydrated

in successive increasing methanol concentrations, incubated overnight with 0.2% $H_2O_2$ in methanol and rehydrated following the reverse protocol. Whole-mount immunofluorescence was performed using a blocking solution composed of 10% DMSO, 0.5% Triton 100X, 2% Bovine Serum Albumine (BSA) and 100 mM glycine, in PBS. Primary and secondary antibodies were applied in the blocking solution. The following primary antibodies were used: anti-GFP rabbit IgG (1/500, Invitrogen Fisher, #A11222), anti-Neurofilament 160 kDa, RMO-270 mouse monoclonal IgG (1/400, Thermofisher #13-0700), anti-HNK1 mouse IgM (1/500, 3H5, DSHB), anti-mitochondria mouse IgG (1/500, milipore #MAB1273) and the following secondary antibodies: Alexa 555 anti-rabbit IgG (1/500, Invitrogen, #A21429), FP-647H anti-rabbit IgG (1/500, Interchim #FP-SC5110), Alexa 555 anti-mouse IgG (1/500, Invitrogen #A31570) and Alexa 647 anti-mouse IgM (1/500, Invitrogen #A21238).

Embryos were then cleared using a modified Ethyl cinnamate (Sigma–Aldrich, #112372) protocol[58]. Embryos were dehydrated in ethanol successive baths before clearing with ECi. Cleared samples were imaged using a light-sheet UltraMicroscope (LaVision Biotech, laser power set at 3–5%, sheet NA 0.035, sheet width 60%).

*Analysis of 3D light sheet images.* 3D images were built and quantified with Arivis Vision4D software. Localization of NB primary tumors and metastases, metastases volume and their distance from primary tumors were determined using an automated image analysis pipeline set up for this project and based on intensity threshold of anti-GFP (IGR-N91 cells) or anti-mito (patient samples) immunolabeling. Sympathetic chain volumes at E9 were measured for the 6 caudal most sympathetic ganglia by using the intensity-based magic hand tool applied on anti-HNK1 immunolabeling.

*Analysis of published bulk RNASeq datasets.* Gene-Set enrichment analyses in published bulk RNASeq data were carried out using the GSEA software [https://www.gsea-msigdb.org/gsea/index.jsp]. Correlation analyses for two gene sets were performed with R2 Genomics Analysis and Visualization platform [http://r2.amc.nl] on two neuroblastoma patient cohorts: Kocak's cohort (GEO ID: GSE45547, n = 649 samples) and Shi and Fischer's cohort (GEO ID: GSE62564; n = 498 samples). Average z-score values were computed over genes within each gene set and used to relate gene signatures based on Fisher's exact test.

*Analysis of published single cell RNASeq datasets.* Raw sequencing data from Dong et al.[3] were processed following the partially published method details, explaining the different UMAP obtained in the present study. The R package Seurat (v4.0.1) was used to calculate the quality control metrics[59]. To filter out low quality cells, we kept all cells with at least 200 detected genes and less than 10% of mitochondrial genes. Doublet cells were removed thanks to the R package DoubletFinder (v2.0.3)[60]. To merge all samples without biasing the analysis with batch effects while preserving the biological variation, we applied Seurat integration[61] and re-computed a clustering based on the corrected matrix. Scores for the expression of gene signatures in single cells were determined using the AddModuleScore function in Seurat with 100 control genes per analyzed gene. The average expression of the gene signature was determined through the subtraction of the control gene sets expression.

*Statistics and reproducibility.* Control and experimental conditions were treated without any distinction in all experiments. For in vitro aggregation and migration/invasion assays, analysis was done in blind. Embryos were allocated to experimental groups randomly.

All measurements were taken from distinct samples. Statistical treatment of the data was performed with Prism 9.0 (GraphPad). For parametric test, both normality and variances homoscedasticity were checked. In case of non-normality or non-homoscedastic variances, non-parametric tests were used. The statistical tests used in each panel are mentioned in the figure legends. The number of independent experiments is indicated in figure legends, experiments were at least repeated three times. Error bars indicate SEM. Significance was defined for p values inferior to 0.05 (*). **, ***, **** in the figures indicate p values <0.01, <0.001, <0.0001 respectively.

**Reporting summary.** Further information on research design is available in the Nature Research Reporting Summary linked to this article.

## Data availability

The following published bulk RNASeq data were used: GSE85047[62], E-GEOD-45547[63], GSE120572[64], E-MTAB-8248[65], GSE65904[66], GSE67066[67], GSE39582[68], GSE102484[69], TCGA_PAAD, GSE62564[70]. ScRNASeq of human tumor samples and embryonic adrenal gland were exploited respectively from Dong et al.[3] (2020; GEO ID: GSE137804; and Jansky et al.[5] (2021; EGA ID: EGAS00001004388;) public datasets. Raw and analyzed bulk RNASeq data generated in this study have been deposited in the Gene Expression Omnibus (GEO) under accession number GSE169280. Proteomic data generated in this study have been deposited to the ProteomeXchange Consortium via the PRIDE partner

repository with the dataset identifier PXD027499. Source data are provided with this paper. The remaining data is available in the Article and in Supplementary Information files.

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

## Acknowledgements

We wish to thank the ProfilExpert core facility (Lyon, France) and especially Joël Lachuer, Severine Croze and Abdel Goumaidi for fruitful assistance in RNASeq data analysis. We thank Dr. Frédérique Dijoud and Isabelle Rochet from the Department of Pathology, Women and Children's Hospital (Bron, France) for the preparation of fresh tumor biopsies. We thank NeuroBioTec, CRB HCL (BB-0033-00046) (Lyon, France) for the assignment of frozen human neuroblastoma samples. We wish to thank Stéphane Chabrier and APICIL for the sponsoring of Oncofactory activities on paediatric cancers. This work has been supported by the Fondation Bettencourt-Schueller (V.C.), the Fondation ARC pour la recherche sur le cancer (PJA 20181207900) (C.D.B.) and by grant from the INCa (PLBIO18-161) (V.C.). This work was conducted within the framework of the LABEX CORTEX and Labex DevWeCAN of Université de Lyon, within the program 'Investissements d'Avenir' (ANR-11-IDEX-0007) operated by the French National Research Agency (ANR) (V.C. and C.D.B.). The proteomic experiments were partially supported by Agence Nationale de la Recherche under projects ProFI (Proteomics French Infrastructure, ANR-10-INBS-08) (Y.C.) and GRAL, a program from the Chemistry Biology Health (CBH) Graduate School of University Grenoble Alpes (ANR-17-EURE-0003) (Y.C.).

## Author contributions

C.D.B. and V.C. designed and supervised the project and the experimental plan. D.B.A. produced conditioned media, performed in vitro and ex vivo experiments, realized the grafts of NB cell lines and the in vivo analysis of OLFM1 Ab. K.T. contributed to the aggregation and migration assays, immunofluorescence on cryosections and in whole embryos, imaging and data analysis. O.I. and B.V. analyzed proteomic and RNASeq data, conceived and analyzed gene signatures in patient cohorts. F.R. performed the Western Blot experiments. C.C. and L.J. contributed to the grafts of patient samples and injection of OLFM1 Ab in vivo. immunohistochemistry, imaging, tumor microdissection, data analysis and molecular biology. B.V. performed in vitro experiments with OLFM1 Ab. J.N.D. and Y.C. realized the mass spectrometry experiments on conditioned media. J.F.B. conceived and set up the antibody against bird's Phox2b protein. V.Co. provided NB cell lines and patient samples. N.C. provided clinical input and patient samples. C.D.B. and V.C. wrote the manuscript.

## Competing interests

The authors declare no competing interests.
