## [Peer Review File · Nature Communications]

Environmental cues from neural crest derivatives act as metastatic triggers in an embryonic neuroblastoma modelREVIEWER COMMENTS

Reviewer #1 (Remarks to the Author):

The manuscript “Non-cell autonomous signaling of nervous gangliogenesis drives neuroblastoma dissemination via a core gene program common to neural crest derived cancers” by Dounia Ben Amar et al dissects exogenous stimuli that promote dissemination of neuroblastoma cells, a hallmark of high-risk neuroblastoma, during embryonic development. To do so the authors use time-resolved proteomics and transcriptomics to study an ex vivo co-cultivation model of neuroblastoma cell lines and explanted chick embryo sympathetic ganglia and dorsal root ganglia as well as an avian patient derived xenograft model. They demonstrate that secreted factors, and among those specifically OLFM1 is released by embryonic sympathetic ganglia mediates the loss of cell-cell contact and promotes migration and a mesenchymal gene expression signature in neuroblastoma cells. The authors also claim that a OLFM1 family expression signature or an independently generated gene set associated with migration is enriched in neuronal crest derived metastatic as opposed to localized tumors.

The study is relevant to better understand neuroblastoma tumor cell dissemination, since patients with disseminated disease have unfavorable prognosis. The work presented is original and novel and is presented in a concise and mostly clear way (some English editing is suggested). The experimental setup and chosen models are appropriate and well controlled. There are however some concerns, especially regarding the nature and relevance of the proposed metastasis-associated gene sets in human tumors and regarding the physiological role of OLFM1 during embryogenesis. In addition a mechanism of action of OLFM1 on neuroblastoma metastasis is lacking.

Specific comments:

Major

1. In line 383 the authors state that recombinant OLFM1 was added in their experimental system to study the effect of OLFM1 in a physiological environment: was human recombinant protein used? The lack of response to OLFM1 by avian cells might be explained by inter-species incompatibility. Therefore it is uncertain whether the conclusion by the authors that “OLFM-mediated anti-cohesive, pro-detachment and pro-migration properties are not shared between NB cells and sympathetic cell contingent, but tumor-specific.” is supported by the data presented.
2. It is not clear how the authors arrived at the gene list they are referring to as “primary tumor cell escape signature”. Is this a list of top-regulated genes and if yes which cell types were compared? Or rather “cherry-picking” the most promising candidates?
3. Presentation of data in figure 7d is confusing. In addition, data seem not very consistent in different data sets and the authors overstate when they claim that “we observed a systematic regulation of both OLFM and primary tumor cell escape signatures in neural crest-derived cancers, in metastatic cases versus localized ones.” First, this is not true in all data sets, though a trend is recognizable. Second, the authors seem to consider the two datasets as interchangeable referring to the same biological processes, which is not the case. Third, why are the enrichment scores positive for localized (stage 1, 2, 3) vs metastatic (stage 4) neuroblastoma, but negative for localized (in situ) vs metastatic melanoma? This needs clarification and revision of the figure/table and text.
4. Discussion: the authors claim to present “OLFM1 glycoprotein, whose activity is integrated into a signaling gene network dedicated to the regulation of NB cell-cell cohesion and motility”, but fail to present this network. Same applies for a “gene module” – is this referring to transcript signatures or to actual genes or OLFM1-induced signaling?
5. In line 539-540 the authors claim that “Our study provides a first example of opportunistic exploitation of physiological signals regulating embryogenesis, very early at the metastatic onset.” This is however not supported by their data since they were not able to demonstrate a physiological role of OLFM1.
6. Based on current literature, there is no evidence that Schwann cell precursors undergo malignant transformation and give rise to neuroblastoma. However they give rise to among other cell types

Schwann cells, which do not represent malignant cells but in contrast re-enact nerve repair-associated processes leading to neuronal differentiation (PMID: 29454705, PMID: 33833454, PMID: 33712610). This should be clarified in the discussion.

7. It would be important to indicate how OLFM1 exerts its function? What is the receptor and suggested down-stream signaling?

8. Introduction: sypatho-adrenal neural crest is a misleading term. Suggest to using the term neural crest (NC) or trunk neuroal crest and sympathoadrenal lineage which derives from NC cells. Also the description of SCPs should be revised. There is no indication of SCPs undergoing malignant transformation so far in the literature.

Minor

1. Some paragraphs are difficult to read and the meaning is not always clear, e.g. lines 84-85. It is recommended to check synthax and spelling.

2. The introduction contains a very extensive description of the results and interpretation. It is suggested to shorten this.

3. It should be indicated in the title that this study is conducted in chicken embryos.

4. Figures: it should be made clear to which comparison the significance test is referring to. It is suggested to indicate p-values.

5. Figure 1a and Supplementary Figure 2a: it is not indicated what kind of staining has been applied to the isolated sympathetic chains and DRG, resp.

6. Figure 2c could be simplified as it is not clear why there are several sections in the graph that actually have the same color.

7. Figure 3d enrichment blots are not convincing for NOR gene set (very few transcripts included). Which gene sets were used here?

8. Standard deviations appear very small, e.g. in figures 1, 4 5 and 6: are technical or biological replicates blotted?

9. Figure 4m: how can the mean volume of metastatic foci be higher than the cumulative volume?

10. Reference to Figure 7d is missing in the text.

11. The table detailing information on the patients used in the PDX model needs proper labeling, numbering and spelling check.

12. Some of the figures, e.g. Figure 7c, suppl figure 5 are actually tables and should be presented as such.

Reviewer #2 (Remarks to the Author):

The manuscript "Non-cell autonomous signaling of nervous gangliogenesis drives neuroblastoma dissemination via a core gene program common to neural crest-derived cancers" describes two different sets of genes that characterize surrounding cells and neuroblastoma cells that reduce cell-cell contacts and thus gain the ability to metastasize. One gene, OLFM1, is followed up with additional experiments that use an antibody which inhibits OLFM1 function.

The authors identify two different sets of genes that modulate the migratory behavior of neuroblastoma cells. It remains unclear how the authors converge on the two gene sets and the genes in each set are not described in detail. There is no indication of what the gene sets characterizes other than being specific for a cell type at a specific developmental stage that is described in the manuscript. It appears that an analysis of the transcriptome of various cell types and developmental stages resulted in these genes as differentially expressed which the authors then assembled into sets. However, methods are not clearly specified and thus the relevance of the sets remains unclear. The term 'gene set' might be misleading here because gene sets are typically based on other information (like literature evidence) that are orthogonal to the specific experimental approach. Here, the newly gene sets appear self-explanatory within the experiments described in the manuscript and thus the results are hermeneutic.

The authors follow up one gene OLFM1 which has been extensively described in the literature. The

literature evidence should be moved from discussion to the introduction in order to clearly indicate the novelty of the findings described here in the manuscript over findings reported previously. Some of the statements are not directly supported by the results such as the existence of an 'olfactomedin gene network composed of interacting partners acting as ligands, receptors and signaling effectors' (l133f). Additional literature or experimental evidence needs to be provided to support such a statement.

Because of a lack of clarity in presentation, it is difficult to clearly point out the novelty of the findings for an uninitiated reader. Specifically, the validity and consistency of the newly defined gene sets needs to be shown or at least justified, and the experimental results of the OLFM1 studies put into a broader context more clearly.

Specific Comments:

The title is unclear. The title sounds circular in which (1) 'non-autonomous signaling of nervous gangliogenesis' (what is 'nervous gangliogenesis?') equals 'a core gene program' and (2) 'neuroblastoma dissemination' equals 'neuronal crest-derived cancers'. Please simplify and use keywords to include all terms that characterize the manuscript and findings but are not mentioned in the title and abstract.

Introduction:

Please write the introduction in plain language. Maybe try to use dependent sentences instead of participle and gerondif.

L68 'NCC' is not explained. In case the second 'C' is used as abbreviation for 'cells' be consistent (eg NC cells -> NCC and NB cell -> NBC).

L112 'tumoral' to 'tumor'. This sentence is overall unclear: 'as physiological counterparts' to what? - be specific.

L120 to 121 please rewrite for clarity.

L121 the sentence does not introduce OLFM1 and why it is important in the this context.

L134 'acting' to 'that are'

Results:

L162f 'pro-disaggregation effect' and 'their "decohesive" properties' is unclear. Please use simple nouns or clearly define the term if used repeatedly.

L193 'transitory' is 'transient'?

L198 consider a different term for 'cohesive mass'?

L205 What are 'factors' and what is a 'generic paraspinal developmental program'?

L227ff What is the source of information that indicated that '12 proteins had already been associated with the regulation of neural crest-related processes'? - Is this literature analysis (citations) or data analysis performed here? If it is a result of the data analysis that is described in the manuscript, please be more specific. This is critical because it remains otherwise unclear why OLFM1 was further analyzed.

L231 'when' (eg 'when confronted') is missing.

L266ff Please explain the criteria that were used to include genes in the 'OLFM-related gene set'). If objective criteria were not used, how did the authors exclude a 'cherry picking' when they assembled

the OLFM-related gene set?

L369 the expression 'highjacked' is unclear. Do NB cells actively modulate OLFM1 signaling?

L371f please move to the introduction. This is relevant background information to OLFM1.

L389 What is 'HNK1' and why was it used?

L392f There is a clear difference in HNK1 staining in the upper part of the tissue section. Please explain or comment. It appears that the interpretation is over-simplified.

L393 Don't understand the term 'tendency' in this context.

L405ff The methods that were used to assemble the 'primary tumor cell escape gene set' remain unclear.

L462-l484 is not discussion but introduction.

L496 please cite the literature when writing 'reported literature'.

L514-532, L605-613 belongs to introduction.

Reviewer #3 (Remarks to the Author):

In this study, the authors use biochemical and imaging tools, together with chick embryo and avian-PDX models to elegantly demonstrate how non cell-autonomous signals from developing peripheral ganglia contribute human NB metastasis. The manuscript builds upon the authors' prior work on human neuroblastoma (NB) metastasis (Delloye-Bourgeois et al., Cancer Cell, 2019). For the most part, the experimental design and findings are robust, and the major claims of the paper are supported by the data. Overall, it is an important and relevant body of work examining a mechanism by which neuroblastoma cells could metastasize. However, there are a few outstanding issues that need to be addressed both experimentally and through changes in the text.

Major concerns:

1. The authors claim in the abstract that “we show that Olfactomedin1 (OLFM1) released by embryonic sympathetic ganglia induces NB cells to shift from noradrenergic to mesenchymal identity, activating a core gene program promoting metastatic onset and dissemination”. This statement is not backed by any experimental evidence that OLFM1 directly induces adrenergic (adr) to mes (mes) transition. The current data (Figs. 3c/d) show that ganglia-conditioned medium results in down and upregulation of adr and mes core regulatory circuitry (CRC) genes respectively, which may or may not be mediated by OLFM1 or related family members. To support their claim, the authors need to show that the CRC is modulated with OLFM1; in other words, that recombinant OLFM1 can induce mes gene expression in adrenergic human cells, and/or that the inhibition of OLFM1 in SG/DRG-conditioned media abrogates the induction of mes CRC genes.

2. The authors mention tumor heterogeneity as a mitigating factor in the varied responses to OLFM1. scRNA-seq data is now available on human adrenal glands at various early developmental stages from the Westermann group (Jansky et al., 2021). Does the gene expression pattern in the human neuroblastoma cell line grown in SG conditioned medium (Fig. 3) correspond to the mes or adr signatures derived from scRNA-seq in the paper by Jansky et al?

3. The authors need to better explain how they arrived at the OLFM-related gene set. Was it derived computationally through methods such as co-expression analysis? Although they mention in the text that “we generated an “OLFM signature” that integrates major OLFM members, their described receptors and direct signaling partners”, there are no details as to which of the 18 genes in the OLFM-signature correspond to the receptors and signaling partners, and the primary references where these gene functions were linked to OLFM1. Without these details, it is difficult to gauge the usefulness of

this signature.

4. Based on the findings in Fig. 6 that blocking OLFM1 function in the chick embryo does not alter the normal developmental program of sympathetic ganglia, the authors claim that “OLFM1 signaling is specifically hijacked by neuroblastoma cells”. While it is true that the data suggest that OLFM1 is functionally relevant to the dissemination of human NB cells and not to normal gangliogenesis, it does not show that OLFM1 signaling is specifically hijacked in NB cells. To demonstrate this, the authors need to experimentally demonstrate how OLFM1 interaction with cognate receptors affects intracellular signaling in NB cells. On that note, it will also be important to explain why only a subset of cells undergo distant metastasis in the IGR-N91 and PDX models, i.e. is there heterogeneity among the tumor cells in terms of expression of genes encoding OLFM1 receptor/downstream signaling components? Analyzing their transcriptomes with and without OLFM1 Ab treatment may provide clues to the underlying mechanism of non-response to the antibody.

5. Although I commend the authors for their effort to pursue the clinical significance of the primary tumor cell escape and OLFM-related signatures, the data is not convincing and lacks consistency in demarcating low-stage, stage 4, and 4S tumors, and other malignant vs benign neural crest-derived tumors. For example, there is no significant enrichment of the escape signature when comparing stages 1/2/3 vs stage 4 in 2 out of 3 NB cohorts (GSE120572, $p=0.11$; E-MTAB-8248, $p=0.06$). Moreover, these cohorts used microarray analysis techniques where the expression of genes are the averaged signal of all tumor cells from the tumor. Unless corroborated by published scRNA datasets, I would suggest that for the purpose of the main figure, the authors should focus on the OLFM1-related signature which seems to perform more consistently (although there are exceptions here as well) and present the remaining data as supplemental information. The overall criticism being that the central message of Fig. 7d is lost in the midst of numbers and the figure should be simplified to highlight the most important message.

6. The Discussion as it stands is overly long and convoluted and reads more like a review article or thesis chapter. It needs to be condensed and focused on a discussion of the salient points. In addition, there are a number of statements that appear to overinterpret the data. Additionally, in the Discussion the authors state that “targeting of OLFM1 achieved in our studies resulted in significant beneficial consequences on the metastatic disease, which suggests that this gene network could represent a promising pathway to consider for future therapies.” The data shown clearly demonstrate that OLFM1 antibodies prevent metastasis; the authors have not shown any data that it treats metastasis. Since we know that neuroblastomas generally present with metastatic disease (and the authors mention this in the Introduction also) how can this prevention strategy affect patients who already present with metastatic disease? Please explain in the discussion, or clarify this further.

6. The paper would be benefited by English editing as there are multiple errors in English usage; some of the sentences end abruptly and it is difficult to understand what the authors mean to convey. Minor comments:

1. The authors used formaldehyde-treated conditioned medium as a negative control to prove that active biomolecules in the conditioned media contribute to its decohesive properties. This is a very crude control. Can the decohesive properties be blocked by conditioned media derived from SG/DRG in the presence of exocytosis inhibitors?

2. In the in vitro studies with the OLFM1 blocking antibody, were the control cells treated with an isotype control? This is not clear from the figure and the methods. It will be important to show in a proof-of-principle assay that the OLFM1 blocking is specific using a species-matched isotype control. For the in vivo studies, what is meant by excipient control? Please clarify.

3. The authors need to validate that IGR-N91 NB cells are adrenergic based on protein expression of known adrenergic (PHOX2B, GATA3, HAND2 etc.), and mesenchymal (PRRX1, FOSL1, RUNX1, SNAI2 etc.) or cite a reference where this has been shown. The original (and follow up) papers that characterized transcriptional states of NB cells (Groningen et al. and Boeva et al., Nat. Genetics, 2017) do not include IGR-N91. Also, the adrn-mes transition using conditioned media and/or recombinant OLFM1 needs to be validated at protein levels using western blot analysis.

4. Figure-3C right panel: the clustering plot is missing part of the hierarchical branch.
5. Please ensure that the nomenclature for sympathetic ganglia and DRG is consistent. For example, there are a few inconsistencies in the label and legend for Fig. 3c.
6. Mouse DRG is either labelled as E15 or E15.5. Please be consistent or clarify if indeed the stages were different between different experiments.

Point-by-point answer to the reviewers

We thank very much the reviewers for their constructive comments. We fully understand that these important points needed to be addressed. In our revisions, we conducted a series of novel experiments and analyses, and deeply modified the manuscript to provide answers to these points. We listed below the above-mentioned modifications.

List of main experiments added to the manuscript:

- All graphs presented as histograms were changed to show individual values
- **Supp Fig 2i**: All candidate proteins highlighted in Fig 2f for their relationship with the neural crest were associated with the corresponding bibliographic references.
- Gene sets were modified according to the reviewers' comments:
 - o **Fig 3g**: OLFM1-related gene set was extended to contain all receptors and direct interactant described in the literature; a table listing all primary references was added as a **Supp. Table 3**.
 - o **Fig 7**: The strategy to extract gene signatures associated with "primary tumor escape" has been modified and precisely explained. Upregulated and downregulated genes have been separated. Refined up and down signatures associated with "metastasis of neural crest-derived cancers" have been extracted from "primary tumor escape" gene sets. Enrichment scores in patient cohorts were presented in a more comprehensive manner. Gene lists are given in a **Supp. Table 5**.
- Data about MES/NOR identities of NB cells were extended:
 - o **Supp Fig 3ab**: the initial NOR identity of IGR-N91 cell line was shown by measuring the expression of key NOR and MES factors both by Western Blot and by qRT-PCR.
 - o **Supp Fig 3c**: The effect of E8-cSG^{cm} on NB cells shift from NOR to MES identity was documented by western blot detection of key MES/NOR proteins in 2 NB cell lines
 - o **Supp Fig 3d**: NB cells shift from NOR to MES identity upon exposure to sympathetic cues was further illustrated by analyzing previously published RNASeq data from our avian model of NB. Upregulation of MES genes and concomitant downregulation of NOR genes in sympathetic tumors as compared to pre-grafted cells was shown.
 - o **Supp Fig 4g**: the involvement of OLFM1 produced by sympathetic ganglia in the NOR to MES shift of NB cells was assessed by measuring the variation of MES genes in NB cells exposed to recombinant OLFM1 or to E8-cSG^{cm} supplemented with OLFM1 Ab.
 - o **Supp Fig 7**: Correlations between genes upregulated in "Primary tumor cell escape" and genes associated with mesenchymal (positive correlation) or noradrenergic (negative correlation) identities were documented. Gene lists associated with these correlative analyses are given in a **Supp. Table 6**.
- **Supp Fig 4a**: endogenous expression of OLFM1 proteins in DRGs and SGs was confirmed by immunofluorescence on chick embryo slices.
- **Supp Fig 4ef**: the specificity of OLFM1 Ab was controlled by using an isotopic control antibody.

- The potential bases explaining heterogeneity in avian-PDX response to OLFM1 blocking antibody were explored in published scRNASeq datasets (**Supp. Fig. 5**):
 - o The OLFM-related gene set was analyzed in Jansky et al. dataset reporting single cell data from human fetal adrenal medullary gland (**Supp. Fig.5a-f**). We could observe a highly heterogeneous expression of the OLFM-related gene set with the different cell identities of the adrenal lineages.
 - o The OLFM-related gene set was analyzed in Dong et al. dataset reporting single cell data from human neuroblastoma tumors (**Supp. Fig.5g-i**). When scoring the OLFM1-gene set in scRNAseq dataset of individual patient tumors and in all tumor cells from the 16 samples, we also found differential representations, which also support functional differences of NB tumors.

- The effect of OLFM1 blocking antibody on sympathetic ganglia was further documented:
 - o **Fig 6ab**: the effect of E8-cSG^{cm} supplemented or not with OLFM1 Ab on dissociated chick sympathetic cells was assessed
 - o **Supp Fig 6ab**: the effect of mouse recombinant OLFM1 on dissociated E15.5 mouse sympathetic cells was assessed
 - o **Fig 6c**: A Phox2b immunofluorescent labeling was performed on E5.5 chick embryo slices to assess the effect of OLFM1 Ab at E3.5 on the differentiation of sympathetic neurons.

- The OLFM1 receptor RTN4R was demonstrated to be involved in sympathetic cues-induced loss of cell-cell cohesion and acquisition of migratory features:
 - o **Fig 7a and 7d**: RTN4R was found upregulated in gene signatures that discriminate metastatic NB from localized stages, and also metastatic neural crest-derived cancers from localized forms.
 - o **Fig 8a-c and Supp Fig8a-c**: Contrary to the OLFM1 receptors APP and GRIA2, RTNR4 high expression was found correlated with bad prognosis in NB patient cohorts
 - o **Fig 8d-j and Supp Fig 8d-l**: Strategies to inhibit RTN4R function (2 siRNAs, one blocking antibody) were shown to block OLFM1- and E8-cSG^{cm}-mediated loss of cell-cell cohesion and induction of migratory behaviors in 2 NB cell lines. No such effect could be observed upon inhibition of APP or GRIA2 expression.

REVIEWER COMMENTS

Reviewer #1 (Remarks to the Author):

The manuscript “Non-cell autonomous signaling of nervous gangliogenesis drives neuroblastoma dissemination via a core gene program common to neural crest derived cancers” by Dounia Ben Amar et al dissects exogenous stimuli that promote dissemination of neuroblastoma cells, a hallmark of high-risk neuroblastoma, during embryonic development. To do so the authors use time-resolved proteomics and transcriptomics to study an ex vivo co-cultivation model of neuroblastoma cell lines and explanted chick embryo sympathetic ganglia and dorsal root ganglia as well as an avian patient derived xenograft model. They demonstrate that secreted factors, and among those specifically OLFM1 is released by embryonic sympathetic ganglia mediates the loss of cell-cell contact and promotes migration and a mesenchymal gene expression signature in neuroblastoma cells. The authors also claim that a OLFM1 family expression signature or an independently generated gene set associated with migration is enriched in neuronal crest derived metastatic as opposed to localized tumors.

The study is relevant to better understand neuroblastoma tumor cell dissemination, since patients with disseminated disease have unfavorable prognosis. The work presented is original and novel and is presented in a concise and mostly clear way (some English editing is suggested). The experimental setup and chosen models are appropriate and well controlled. There are however some concerns, especially regarding the nature and relevance of the proposed metastasis-associated gene sets in human tumors and regarding the physiological role of OLFM1 during embryogenesis. In addition a mechanism of action of OLFM1 on neuroblastoma metastasis is lacking.

Specific comments:

Major

1. In line 383 the authors state that recombinant OLFM1 was added in their experimental system to study the effect of OLFM1 in a physiological environment: was human recombinant protein used? The lack of response to OLFM1 by avian cells might be explained by inter-species incompatibility. Therefore it is uncertain whether the conclusion by the authors that “OLF1-mediated anti-cohesive, pro-detachment and pro-migration properties are not shared between NB cells and sympathetic cell contingent, but tumor-specific.” is supported by the data presented.

We thank the reviewer for this remark and addressed this point by performing two additional sets of experiments. Indeed, recombinant OLFM1 used in the study is produced from mouse, raising the question of inter-species incompatibility when used on avian cells. Thus, we tested recombinant OLFM1 on dissociated E15.5 mouse sympathetic ganglia cells and obtained similar results than on chick sympathetic cells (Supp Fig 6ab). We also tested the effect of chick E8-SG^{cm} supplemented or not with an OLFM1 blocking antibody (previously functionally validated in chick tissues, in Lencinas et al 2013) on dissociated chick sympathetic cells (now Fig 6ab) and could confirm lack of response by avian cells to OLFM1 modulations.

2. It is not clear how the authors arrived at the gene list they are referring to as “primary tumor cell escape signature”. Is this a list of top-regulated genes and if yes which cell types were compared? Or rather “cherry-picking” the most promising candidates?

We are sorry for the lack of clarity regarding the method to select the gene list referring to as “Primary tumor cell escape signature”. We took into account the remarks of the three referees and modified our strategy to select those genes in a more comprehensive and unbiased manner. From our RNASeq data, we worked on the 308 transcripts found differentially and commonly regulated (up or down) between control-treated NB cells and E8-cSG^{cm}/E8-cDRG^{cm}/E15.5-mSG^{cm} (Fig. 3b). These 308 transcripts were separated into “upregulated” and “downregulated” transcripts upon exposure to embryonic cues. Then, using GSEA algorithms, we selected the minimal upregulated gene set allowing to significantly discriminate stage 4 NBs from locoregional stages (1-2-3) (NES > 1.5; FDR < 0.25; p-val < 0.05), in two different published cohorts. The same strategy was performed regarding the downregulated genes (new Fig 7a-c). The relevance of these up & down gene sets was further confirmed by running similar stage 4 NB versus stages 1-2-3 NB comparisons in two other independent cohorts (that have not been used for the selection of the gene sets).

The same type of strategy was used to extract a core gene signature that not only distinguishes stage 4 NBs from locoregional ones, but also more generally discriminates metastatic forms of neural crest-derived cancers from localized ones. To do so, again, we used GSEA algorithms to extract the minimal up and down gene sets (within up and down “Primary tumor cell escape” gene sets) allowing to significantly distinguish metastatic melanomas from local forms in a published and well annotated cohort. The relevance of these up & down core gene sets was further confirmed by running comparisons between malignant and benign forms of pheochromocytomas, another neural crest-derived cancer (independent cohort). These new data are presented in Fig 7d-f.

3. Presentation of data in figure 7d is confusing. In addition, data seem not very consistent in different data sets and the authors overstate when they claim that “we observed a systematic regulation of both OLFM and primary tumor cell escape signatures in neural crest-derived cancers, in metastatic cases versus localized ones.” First, this is not true in all data sets, though a trend is recognizable. Second, the authors seem to consider the two datasets as interchangeable referring to the same biological processes, which is not the case. Third, why are the enrichment scores positive for localized (stage 1, 2, 3) vs metastatic (stage 4) neuroblastoma, but negative for localized (in situ) vs metastatic melanoma? This needs clarification and revision of the figure/table and text.

We fully agree with the reviewer’s comment and completely modified both the data and the way to present them. First, regarding our study on patient cohorts, we focused our analyses on “Primary tumor cell escape” and

“Metastasis of neural crest-derived cancers” gene sets, as we agree that the OLFM1-related gene set refers to a different and transient biological process, that is addressed differently in our study. Second, to be able to assign a biological significance to differences observed between metastatic and localized tumors, we chose to separate up and down gene sets, and to calculate enrichment scores for each of these gene sets separately. We now obtain more consistent and significant scores in each presented comparison. Finally, we decided not to include the melanoma cohort GSE7553, as the number of “in situ” cases was too low (n=2) to run statistical, convincing, analyses. The “table” presentation of these analyses has also been modified and converted to a graphical presentation for more clarity.

4. Discussion: the authors claim to present “OLFM1 glycoprotein, whose activity is integrated into a signaling gene network dedicated to the regulation of NB cell-cell cohesion and motility”, but fail to present this network. Same applies for a “gene module” – is this referring to transcript signatures or to actual genes or OLFM1-induced signaling?

We understand that the terms chosen to describe OLFM1 and its described signaling partners (“signaling network”, “gene module”) were not appropriated. We simplified this part of the discussion, and rather present OLFM1 as being part of a “complex network of signaling molecules functionally linked to OLFM proteins and their context-dependent regulations”. This assumption is not shown in our study but refers to the literature about OLFM signaling in a variety of biological contexts. Indeed, precise data about OLFM1-induced signaling and its molecular partners are still lacking.

5. In line 539-540 the authors claim that “Our study provides a first example of opportunistic exploitation of physiological signals regulating embryogenesis, very early at the metastatic onset.” This is however not supported by their data since they were not able to demonstrate a physiological role of OLFM1.

We fully understand this concern and agree that our statement lacked clarity. Indeed, our intention was to outline that the responses of NB cells to gain and loss of OLFM1 were not shared with sympathoblasts, supporting these responses are tumor-specific. Using HNK1 and Phox2B markers, we further analyzed the morphology of the sympathoadrenal derivatives, after injection of the anti-OLFM1 antibody and found no evidence for cell disaggregation or any other alteration of these structures (Fig 6). Similarly, *in vitro*, aggregated sympathoblasts did not react to OLFM1 exposure, in contrast to NBs that lost their cell-cell cohesion. As indicated in the manuscript, an early role of OLFM1 on pre-migratory NC has been reported but its later contribution in the course of sympatho-adrenal differentiation remains unknown. We focused our revisions on the pathological role of OLFM1, as addressing its physiological functions would require a dedicated experimental program beyond the scope of the present work.

We modified our conclusion, focusing on the differences of behaviors between NB cells and sympathoblasts.

6. Based on current literature, there is no evidence that Schwann cell precursors undergo malignant transformation and give rise to neuroblastoma. However they give rise to among other cell types Schwann cells, which do not represent malignant cells but in contrast re-enact nerve repair-associated processes leading to neuronal differentiation (PMID: 29454705, PMID: 33833454, PMID: 33712610). This should be clarified in the discussion.

We now make it clearer the lack of evidence for a Schwann cell origin of NB.

7. It would be important to indicate how OLFM1 exerts its function? What is the receptor and suggested downstream signaling?

The mode of action of OLFM1 was left fully open in our initial work. We worked out to address this issue (new Fig 8 and Supp. Fig 8).

Three membrane proteins were mainly reported in the literature to contribute to OLFM functions (APP, GRIA2/AMPA Receptor and RTN4R/NOGO Receptor). We found all are expressed at the transcriptional level by NB cell lines and in NB patient cohorts. Using 2 different siRNA and a function-blocking antibody, we could show that RTN4R is mandatory for OLFM1-triggered regulation of NB cell-cell cohesion and migration. In contrast, altering the expression of the two others in NB cells had no effect.

We also provide a correlative analysis of OLFM1 putative receptors expression levels and overall survival in 2 NB patient cohorts. Remarkably, it revealed a robust negative correlation for RTN4R but not the other receptors, which fully supports our findings that this receptor mediates the pro-metastatic effects of OLFM1.

8. Introduction: sympatho-adrenal neural crest is a misleading term. Suggest to using the term neural crest (NC) or trunk neuroal crest and sympathoadrenal lineage which derives from NC cells. Also the description of SCPs should be revised. There is no indication of SCPs undergoing malignant transformation so far in the literature.

This has been amended in the revised manuscript.

Minor

1. Some paragraphs are difficult to read and the meaning is not always clear, e.g. lines 84-85. It is recommended to check synthax and spelling.

Done

2. The introduction contains a very extensive description of the results and interpretation. It is suggested to shorten this.

Done

3. It should be indicated in the title that this study is conducted in chicken embryos.

Done

4. Figures: it should be made clear to which comparison the significance test is referring to. It is suggested to indicate p-values.

We modified the legends and/or the figures to describe more clearly the comparison performed in significance tests. The legends of the p-values are indicated in the methods section as it is the same throughout the main figures.

5. Figure 1a and Supplementary Figure 2a: it is not indicated what kind of staining has been applied to the isolated sympathetic chains and DRG, resp.

Indeed, no staining was applied to isolated SGs and DRGs, the presented images are photos taken with white light with a stereomicroscope. This has been indicated in the legends of the corresponding Figures.

6. Figure 2c could be simplified as it is not clear why there are several sections in the graph that actually have the same color.

We modified Figure 2c to group sections that have the same color: each section is indeed related to a single GO term (biological process), and the color relate to a group of GO terms (level 2) affiliated to the same level 1 GO term.

7. Figure 3d enrichment blots are not convincing for NOR gene set (very few transcripts included). Which gene sets were used here?

We used the NOR gene set described by Boeva et al. (2017), enriched with a few genes commonly found in the literature describing NOR/ADR signatures (Van Groningen et al. 2017; Jansky et al. 2021; Dong et al. 2020). We also tested an exhaustive NOR/ADR gene set (NOR-all, presented in Supp. Fig 7 and Supp. table 6) that led to the same type of result. We assume that the NOR to MES shift described upon exposure to embryonic cues is dynamic and transient and might be more clearly observed in the upregulation of MES-related genes than in the downregulation of a well-established NOR initial status in IGR-N91 cells. However, we could further document this shift both by Western Blot and qRT-PCR on NOR and MES key factors (Supp Fig 3bc).

8. Standard deviations appear very small, e.g. in figures 1, 4 5 and 6: are technical or biological replicates blotted? Error bars show SEM in all the graph presented. For cell aggregation experiments where SEM appear small, biological replicates are blotted, each aggregate being considered as a replicate, and the whole experiment being repeated at least 3 times.

9. Figure 4m: how can the mean volume of metastatic foci be higher than the cumulative volume?

The mean volume of metastatic foci is indeed lower than the cumulative volume: this impression is due to the different scales in the same graph (10^5 versus $10^7 \mu\text{m}^3$).

10. Reference to Figure 7d is missing in the text.

Figure 7d is now Fig 7c and 7f and the missing reference has been added in the text.

11. The table detailing information on the patients used in the PDX model needs proper labeling, numbering and spelling check.

We are not sure what the reviewer means here. We extracted the table detailing information on the patients from the Material and Methods section and numbered it as a Supplementary Table 4. We hope this is what was expected here.

12. Some of the figures, e.g. Figure 7c, suppl figure 5 are actually tables and should be presented as such. Fig 7c and suppl fig 5 have been modified and are no longer presented as tables. List of genes and biological processes have been presented as Tables.

Reviewer #2 (Remarks to the Author):

The manuscript "Non-cell autonomous signaling of nervous gangliogenesis drives neuroblastoma dissemination via a core gene program common to neural crest-derived cancers" describes two different sets of genes that characterize surrounding cells and neuroblastoma cells that reduce cell-cell contacts and thus gain the ability to metastasize. One gene, OLFM1, is followed up with additional experiments that use an antibody which inhibits OLFM1 function.

The authors identify two different sets of genes that modulate the migratory behavior of neuroblastoma cells. It remains unclear how the authors converge on the two gene sets and the genes in each set are not described in detail. There is no indication of what the gene sets characterize other than being specific for a cell type at a specific developmental stage that is described in the manuscript. It appears that an analysis of the transcriptome of various cell types and developmental stages resulted in these genes as differentially expressed which the authors then assembled into sets. However, methods are not clearly specified and thus the relevance of the sets remains unclear. The term 'gene set' might be misleading here because gene sets are typically based on other information (like literature evidence) that are orthogonal to the specific experimental approach. Here, the newly gene sets appear self-explanatory within the experiments described in the manuscript and thus the results are hermeneutic.

The authors follow up one gene OLFM1 which has been extensively described in the literature. The literature evidence should be moved from discussion to the introduction in order to clearly indicate the novelty of the findings described here in the manuscript over findings reported previously.

Some of the statements are not directly supported by the results such as the existence of an 'olfactomedin gene network composed of interacting partners acting as ligands, receptors and signaling effectors' (l133f). Additional literature or experimental evidence needs to be provided to support such a statement.

Because of a lack of clarity in presentation, it is difficult to clearly point out the novelty of the findings for an uninitiated reader. Specifically, the validity and consistency of the newly defined gene sets needs to be shown or at least justified, and the experimental results of the OLFM1 studies put into a broader context more clearly.

Specific Comments:

The title is unclear. The title sounds circular in which (1) 'non-autonomous signaling of nervous gangliogenesis' (what is 'nervous gangliogenesis?') equals 'a core gene program' and (2) 'neuroblastoma dissemination' equals 'neural crest-derived cancers'. Please simplify and use keywords to include all terms that characterize the manuscript and findings but are not mentioned in the title and abstract.

We changed the title according to the referee's recommendations and Nature Communications guidelines.

Introduction:

Please write the introduction in plain language. Maybe try to use dependent sentences instead of participles and gerunds.

We deeply modified and reorganized the introduction, trying to integrate comments from all referees.

L68 'NCC' is not explained. In case the second 'C' is used as abbreviation for 'cells' be consistent (eg NC cells -> NCC and NB cell -> NBC).

done

L112 'tumoral' to 'tumor'. This sentence is overall unclear: 'as physiological counterparts' to what? - be specific.

done

L120 to 121 please rewrite for clarity.

done

L121 the sentence does not introduce OLFM1 and why it is important in the this context.

done

L134 'acting' to 'that are'

done

Results:

L162f 'pro-disaggregation effect' and 'their “decohesive” properties' is unclear. Please use simple nouns or clearly define the term if used repeatedly.

We now adopted a unique term: loss of cell-cell cohesion (or reduction of cell-cell cohesion) to be more consistent.

L193 'transitory' is 'transient'?

change made

L198 consider a different term for 'cohesive mass'?

We change the term by “adhesive aggregate”.

L205 What are 'factors' and what is a 'generic paraspinal developmental program'?

Our intention was to underline developmental processes shared by DRGs and SGs that are both neural-crest derivatives. We deleted the term ‘generic paraspinal developmental program’ and explain our intention more clearly.

The term “factors” was inappropriately used, we replaced it in the text by “cues” or “proteins” accordingly.

L227ff What is the source of information that indicated that '12 proteins had already been associated with the regulation of neural crest-related processes'? - Is this literature analysis (citations) or data analysis performed here? If it is a result of the data analysis that is described in the manuscript, please be more specific. This is critical because it remains otherwise unclear why OLFM1 was further analyzed.

The above-mentioned sentence “12 proteins had already been associated with the regulation of neural-crest-related processes” is the result of literature analysis. We have now listed the corresponding references in Supp. Fig. 2i.

L231 'when' (eg 'when confronted') is missing.

done

L266ff Please explain the criteria that were used to include genes in the 'OLFM-related gene set'. If objective criteria were not used, how did the authors exclude a 'cherry picking' when they assembled the OLFM-related gene set?

To include genes in the OLFM-related gene set, we performed an intensive review of the currently available literature about OLFM proteins, their described direct transmembrane receptors/interactants, and signaling partners directly linked to OLFM/receptor interactions and expressed in our RNASeq data. Of course, we can't argue for an exhaustive gene set as the literature about OLFM proteins is still scarce. Meanwhile, with these criteria, we extended our first version of OLFM-related gene set to be as objective as possible.

L369 the expression 'highjacked' is unclear. Do NB cells actively modulate OLFM1 signaling?

We modify the text, describing that NB cells “use” or “utilize” physiological signals.

L371f please move to the introduction. This is relevant background information to OLFM1.

done

L389 What is 'HNK1' and why was it used?

The information has now been provided more clearly in the text.

L392f There is a clear difference in HNK1 staining in the upper part of the tissue section. Please explain or comment. It appears that the interpretation is over-simplified.

We apologize for the difference appearing in the provided illustrations. It was indeed due to different antero-posterior level of the showed transverse sections. We now went to a quantitative analysis of the HNK1 immunolabeling, which show unambiguously that anti-OLFM1 injection has no impact on these structures. We

also analyzed the profile of Phox2B, a key gene of sympathetic differentiation and as observed for HNK1, found no effect of the anti-OLMF1 injection.

L393 Don't understand the term 'tendency' in this context.

As detailed in the next point, we now deeply rethought the strategy of bioinformatic analysis and the signatures provided in the revised manuscript significantly distinguish metastatic stages from other stages, both for NB and other NC-derived cancer, which is not the case for cancers unrelated to the NC.

L405ff The methods that were used to assemble the 'primary tumor cell escape gene set' remain unclear.

We are sorry for the lack of clarity regarding the method to select the gene list referring to as "Primary tumor cell escape signature". We took into account the remarks of the three referees and modified our strategy to select those genes in a more comprehensive and unbiased manner. From our RNASeq data, we worked on the 308 transcripts found differentially and commonly regulated (up or down) between control-treated NB cells and E8-cSG^{cm}/E8-cDRG^{cm}/E15.5-mSG^{cm} (Fig. 3b). These 308 transcripts were separated into "upregulated" and "downregulated" transcripts upon exposure to embryonic cues. Then, using GSEA algorithms, we selected the minimal upregulated gene set allowing to significantly discriminate stage 4 NBs from locoregional stages (1-2-3) (NES > 1.5; FDR < 0.25; p-val < 0.05), in two different published cohorts. The same strategy was performed regarding the downregulated genes (new Fig 7a-c). The relevance of these up & down gene sets was further confirmed by running similar stage 4 NB versus stages 1-2-3 NB comparisons in two other independent cohorts (that have not been used for the selection of the gene sets).

The same type of strategy was used to extract a core gene signature that not only distinguishes stage 4 NBs from locoregional ones, but also more generally discriminates metastatic forms of neural crest-derived cancers from localized ones. To do so, again, we used GSEA algorithms to extract the minimal up and down gene sets (within up and down "Primary tumor cell escape" gene sets) allowing to significantly distinguish metastatic melanomas from local forms in a published and well annotated cohort. The relevance of these up & down core gene sets was further confirmed by running comparisons between malignant and benign forms of pheochromocytomas, another neural crest-derived cancer (independent cohort). These new data are presented in Fig 7d-f.

L462-l484 is not discussion but introduction.

The sentences have been moved to the introduction.

L496 please cite the literature when writing 'reported literature'.

Done

L514-532, L605-613 belongs to introduction.

The sentences have been moved to the introduction.

Reviewer #3 (Remarks to the Author):

In this study, the authors use biochemical and imaging tools, together with chick embryo and avian-PDX models to elegantly demonstrate how non cell-autonomous signals from developing peripheral ganglia contribute human NB metastasis. The manuscript builds upon the authors' prior work on human neuroblastoma (NB) metastasis (Delloye-Bourgeois et al., Cancer Cell, 2017). For the most part, the experimental design and findings are robust, and the major claims of the paper are supported by the data. Overall, it is an important and relevant body of work examining a mechanism by which neuroblastoma cells could metastasize. However, there are a few outstanding issues that need to be addressed both experimentally and through changes in the text.

Major concerns:

1. The authors claim in the abstract that "we show that Olfactomedin1 (OLFM1) released by embryonic sympathetic ganglia induces NB cells to shift from noradrenergic to mesenchymal identity, activating a core gene program promoting metastatic onset and dissemination". This statement is not backed by any experimental evidence that OLFM1 directly induces adrenergic (adr) to mes (mes) transition. The current data (Figs. 3c/d) show that ganglia-conditioned medium results in down and upregulation of adr and mes core regulatory circuitry (CRC)

genes respectively, which may or may not be mediated by OLFM1 or related family members. To support their claim, the authors need to show that the CRC is modulated with OLFM1; in other words, that recombinant OLFM1 can induce mes gene expression in adrenergic human cells, and/or that the inhibition of OLFM1 in SG/DRG-conditioned media abrogates the induction of mes CRC genes.

We agree with the referee's point and worked on additional experiments to document the effect of OLFM1 on the expression of MES genes. We quantified the expression of key MES genes (JUN, MAML2, RUNX1) in two different neuroblastoma cell lines with known initial noradrenergic identity (SH-SY5Y – Boeva et al., 2017 and IGR-N91 – this paper), upon recombinant OLFM1 or E8-cSG^{cm} exposure. We could show that adding either recombinant OLFM1 or E8-cSG^{cm} to NB cell lines triggered an increase in the expression of JUN, MAML2 or RUNX1 genes. Notably, the addition of OLFM1 blocking antibody to E8-cSG^{cm} at least partially blocked E8-cSG^{cm}-induced increase in the expression of MES genes (Supp. Fig. 4g). Hence, OLFM1 appears to be, at least in part, involved in the acquisition of mesenchymal features by NB cells.

2. The authors mention tumor heterogeneity as a mitigating factor in the varied responses to OLFM1. scRNA-seq data is now available on human adrenal glands at various early developmental stages from the Westermann group (Jansky et al., 2021). Does the gene expression pattern in the human neuroblastoma cell line grown in SG conditioned medium (Fig. 3) correspond to the mes or adr signatures derived from scRNA-seq in the paper by Jansky et al?

We thank the referee for his/her suggestion and added supplementary analyses to our work accordingly. Here, the referee questioned the heterogeneous OLFM1-triggered response of patient tumoral cells. This heterogeneity might be either the result of differences of cell of origin or of tumoral events.

First, we analyzed the expression of OLFM-related gene set as a whole but also for each individual gene of the gene set in scRNASeq data of human adrenal gland at different developmental stages (Jansky et al., 2021) (new Supp. Fig5a-f). We observed a strong heterogeneity of expression for the whole gene set in all the annotated cell types of the fetal adrenal gland. This heterogeneity was even more pronounced at an early developmental stage (8 weeks post-conception). Thus, it seems that the OLFM1 signature does not distinguish a particular cell cluster of the developing human adrenal gland. Rather it distributes in the whole cell population. Nevertheless, we found differential representation and expression levels of individual genes composing the signature among the cell identities, which provides a basis for functional differences in response to OLFM1.

Furthermore, such a heterogeneity was also observed in scRNASeq data of 16 NB tumors published by Dong et al (2020). This result was obtained considering all tumor cells from the 16 NBs (Supp Fig. 5h) and also between each tumor (Supp Fig. 5i), reinforcing the idea that NB heterogeneity regarding OLFM1-gene set expression could participate in the varied responses to OLFM1 blocking antibody. This argument is further developed in point 4.

Regarding the second part of the question, we could analyze the expression of genes upregulated in "primary tumor cell escape" in single cell RNASeq data of NB tumors (from Dong et al). Interestingly, we observed that the expression of the signature we identified here was strongly increased in NB cells assigned with a mesenchymal identity (Supp Fig. 7e-g).

3. The authors need to better explain how they arrived at the OLFM-related gene set. Was it derived computationally through methods such as co-expression analysis? Although they mention in the text that "we generated an "OLFM signature" that integrates major OLFM members, their described receptors and direct signaling partners", there are no details as to which of the 18 genes in the OLFM-signature correspond to the receptors and signaling partners, and the primary references where these gene functions were linked to OLFM1. Without these details, it is difficult to gauge the usefulness of this signature.

We apologize for the lack of clarity regarding the building of OLFM-related gene set. To include genes in the OLFM-related gene set, we performed an intensive review of the currently available literature about OLFM proteins, their described direct transmembrane receptors/interactants, and signaling partners directly linked to OLFM/receptor interactions and expressed in our RNASeq data. Of course, we can't argue for an exhaustive gene set as the literature about OLFM proteins is still scarce. Meanwhile, with these criteria, we extended our first version of OLFM-related gene set to be as objective as possible. We also added a Supplementary table (Supp. Table 3) listing the primary references for each member of gene set.

4. Based on the findings in Fig. 6 that blocking OLFM1 function in the chick embryo does not alter the normal developmental program of sympathetic ganglia, the authors claim that “OLFM1 signaling is specifically hijacked by neuroblastoma cells”. While it is true that the data suggest that OLFM1 is functionally relevant to the dissemination of human NB cells and not to normal gangliogenesis, it does not show that OLFM1 signaling is specifically hijacked in NB cells. To demonstrate this, the authors need to experimentally demonstrate how OLFM1 interaction with cognate receptors affects intracellular signaling in NB cells. On that note, it will also be important to explain why only a subset of cells undergo distant metastasis in the IGR-N91 and PDX models, i.e. is there heterogeneity among the tumor cells in terms of expression of genes encoding OLFM1 receptor/downstream signaling components? Analyzing their transcriptomes with and without OLFM1 Ab treatment may provide clues to the underlying mechanism of non-response to the antibody.

We fully agree that our statement lacked clarity. As the referee understood it, our intention was to outline that the responses of NB cells to gain and loss of OLFM1 were not shared with sympathoblasts, supporting these responses are tumor-specific. The mode of action of OLFM1 on NB cells was left fully open in our initial work. We worked out to address this issue, starting from the literature that reported three membrane proteins contributing to OLFM functions (APP, GRIA2/AMPA Receptor and RTN4R/NOGO Receptor).

Notably RTN4R was the only one to be found out among the upregulated genes of the “primary tumor cell escape” and “metastasis of neural crest-related cancers” signatures (Fig 7a and 7d).

We found all are expressed at the transcriptional level by NB cell lines and in NB patient cohorts. Using 2 different siRNA and a function-blocking antibody, we could show that RTN4R is mandatory for OLFM1-triggered regulation of NB cell-cell cohesion and migration. In contrast, altering the expression of the two others in NB cells had no effect (Fig 8 and Supp Fig 8).

We also provide a correlative analysis of OLFM1 putative receptors expression levels and overall survival in 2 NB patient cohorts. Remarkably, it revealed a robust negative correlation for RTN4R but not the other receptors, which fully supports our findings that this receptor mediates the pro-metastatic effects of OLFM1 (Fig 8 and Supp Fig 8).

We hope that the referee will agree that further elucidating the downstream pathways would require a specific program that is beyond the scope of the present grounding work.

As the referee mentioned it, only some cells escaped from the primary tumor and this was counteracted by OLFM1 blockade. This indicates that not all NB cells might react to OLFM1 exposure. Moreover, it is also likely that the temporality of their primary tumor escape also show differences among disseminating cells. This heterogeneity might be grounded by a number of features.

Our bioinformatic data revealed a gene network composed of OLFMs and their interactants, that is significantly regulated upon exposure to embryonic signals. The outcome of OLFM1 exposure at the individual cell levels might depend on the global activity of the network in each cell, representativity of individual components, as well as it might rely on cell-type specificities. To document heterogeneity features, we thought to evaluate the status of the OLFM1 signature in individual cells, taking advantage of the single cell RNAseq data set of NB patient tumors recently made public (Dong et al, 2020). Our analysis revealed differential scores between patient tumor samples, which attests for heterogeneity among NB patients. RTN4R on its own is unlikely to predict how a cell will react to OLFM1 but nevertheless its expression is mandatory. In patient tumors, RTN4R had heterogeneous profile, which also indicates that some but not all NB cells are equipped for responding to OLFM1 (**Supp. Fig 8m**).

5. Although I commend the authors for their effort to pursue the clinical significance of the primary tumor cell escape and OLFM-related signatures, the data is not convincing and lacks consistency in demarcating low-stage, stage 4, and 4S tumors, and other malignant vs benign neural crest-derived tumors. For example, there is no significant enrichment of the escape signature when comparing stages 1/2/3 vs stage 4 in 2 out of 3 NB cohorts (GSE120572, $p=0.11$; E-MTAB-8248, $p=0.06$). Moreover, these cohorts used microarray analysis techniques where the expression of genes are the averaged signal of all tumor cells from the tumor. Unless corroborated by published scRNA datasets, I would suggest that for the purpose of the main figure, the authors should focus on the OLFM1-related signature which seems to perform more consistently (although there are exceptions here as well) and present the remaining data as supplemental information. The overall criticism being that the central message of Fig. 7d is lost in the midst of numbers and the figure should be simplified to highlight the most important message.

We fully agree with the referee’s comment and made significant changes to provide more consistent and clearer data. As explained above, we first modified our strategy to extract our “Primary tumor cell escape” and a novel

“Metastasis of neural crest-derived cancers” gene sets, both being now driven by data from patient cohorts. Second, to be able to assign a biological significance to differences observed between metastatic and localized tumors, we chose to separate up and down gene sets, and to calculate enrichment scores for each of these gene sets separately. For each cancer cohort, we focused our message on the comparison between stage 4/metastatic and locoregional/benign tumors, and left cohorts for which the number of cases was too low to run statistical, convincing analyses (GSE7553). We now obtain more consistent and significant scores in each presented comparison. The “table” presentation of these analyses has also been modified and converted to a graphical presentation to highlight the most important message. We hope that these profound modifications will fit with the referee’s expectations.

6. The Discussion as it stands is overly long and convoluted and reads more like a review article or thesis chapter. It needs to be condensed and focused on a discussion of the salient points. In addition, there are a number of statements that appear to overinterpret the data. Additionally, in the Discussion the authors state that “targeting of OLFM1 achieved in our studies resulted in significant beneficial consequences on the metastatic disease, which suggests that this gene network could represent a promising pathway to consider for future therapies.” The data shown clearly demonstrate that OLFM1 antibodies prevent metastasis; the authors have not shown any data that it treats metastasis. Since we know that neuroblastomas generally present with metastatic disease (and the authors mention this in the Introduction also) how can this prevention strategy affect patients who already present with metastatic disease? Please explain in the discussion, or clarify this further.

We now cut the discussion and moved several paragraphs to the introduction.

We fully agree with this comment and modified the discussion accordingly by adding “Although metastatic stages are already manifested at the diagnosis (Maris, 2010) interfering with OLFM1 could slow-down the progression of disseminating NB cells.

6. The paper would be benefited by English editing as there are multiple errors in English usage; some of the sentences end abruptly and it is difficult to understand what the authors mean to convey.

We hope the deep editing made in the text improved the writing and will facilitate the reading.

Minor comments:

1. The authors used formaldehyde-treated conditioned medium as a negative control to prove that active biomolecules in the conditioned media contribute to its decohesive properties. This is a very crude control. Can the decohesive properties be blocked by conditioned media derived from SG/DRG in the presence of exocytosis inhibitors?

We agree with the referee that the use of formaldehyde-treated conditioned medium as a negative control is crude. We tried to use exocytosis inhibitors (Exo1, Brefeldin A) in our conditioned media. However, exocytosis inhibitors are rather designed to be used in short time course, a few minutes at most. It is far from our experimental protocol of conditioned media production that implies the culturing of SG/DRG living tissues for 48h to obtain sufficient amounts of material. The presence of Exo1/Brefeldin A in our SG/DRGs cultures for 48 hours was indeed toxic for the tissues, we faced strong tissue damages that made not possible the use of the conditioned medium.

2. In the in vitro studies with the OLFM1 blocking antibody, were the control cells treated with an isotype control? This is not clear from the figure and the methods. It will be important to show in a proof-of-principle assay that the OLFM1 blocking is specific using a species-matched isotype control. For the in vivo studies, what is meant by excipient control? Please clarify.

We performed additional experiments using a control isotypic antibody (Sheep IgG) to better assess the specificity of action of OLFM1 blocking antibody (Supp. Fig. 4ef) in cell aggregation assays. For the in vivo studies, the control used was the excipient of OLFM1 blocking antibody, -ie: PBS- this was clarified in the methods section.

3. The authors need to validate that IGR-N91 NB cells are adrenergic based on protein expression of known adrenergic (PHOX2B, GATA3, HAND2 etc.), and mesenchymal (PRRX1, FOSL1, RUNX1, SNAI2 etc.) or cite a reference where this has been shown. The original (and follow up) papers that characterized transcriptional states of NB cells (Groningen et al. and Boeva et al., Nat. Genetics, 2017) do not include IGR-N91. Also, the

adrn-mes transition using conditioned media and/or recombinant OLFM1 needs to be validated at protein levels using western blot analysis.

We now have more formally characterized the adrenergic status of IGR-N91 cell line by analyzing the expression levels of key NOR and MES genes both at the transcriptional level and at the protein level (Supp. Fig.3ab). We compared these expressions with those of previously characterized cell lines: the mesenchymal SHEP cell line and the adrenergic SH-SY5Y cell line (Boeva et al. 2017).

Furthermore, we now validated the NOR-to-MES transition upon cSG^{cm} exposure by western blot analysis of core NOR and MEs proteins in two different NB cell lines (IGR-N91 and SH-SY5Y) (Supp. Fig 3c). These data were further reinforced by analyzing the induction of MES genes upon exposure to recombinant OLFM1 or cSG^{cm} (Supp. Fig. 4g). The upregulation of MES genes upon cSG^{cm} exposure was at least partially blocked when an OLFM1 blocking antibody was added, suggesting a direct involvement of OLFM1 in the NOR-to-MES transition.

4. Figure-3C right panel: the clustering plot is missing part of the hierarchical branch.

We corrected this point in Fig3C.

5. Please ensure that the nomenclature for sympathetic ganglia and DRG is consistent. For example, there are a few inconsistencies in the label and legend for Fig. 3c.

Done

6. Mouse DRG is either labelled as E15 or E15.5. Please be consistent or clarify if indeed the stages were different between different experiments.

E15.5 is the right stage for all experiments; we modified it accordingly.

REVIEWER COMMENTS

Reviewer #1 (Remarks to the Author):

Comments to Dounia Ben Amar et al – revised manuscript

The authors now provide new data and have substantially revised the manuscript in response to the concerns and comments raised by me and the other reviewers. The manuscript has substantially improved, but I have still concerns regarding the interpretation of the newly added analysis of bulk and single cell RNA-sequencing datasets. Furthermore, the manuscript needs some further revision, especially the abstract and results section and it is recommended to condense the figures in order to present a coherent and concise storyline.

1. The title of the manuscript shall make clear that embryonic neural crest derivatives is referring to cells.
2. The abstract should be revised to present the main findings in a clear way.
3. The language and specific terms used are sometimes not entirely clear and accurate. Please revise the manuscript throughout with a focus on precise and concise language.
4. While the effort the authors put into presenting additional new data in Figure 7 and other figures is appreciated as they provide new insights and improve the manuscript, the results section is now overly long and should be condensed. It is suggested to reduce the figures to present only the core relevant data as main figures. E.g. Fig 2c does not add much information, Fig. 2e could be presented in a condensed form or moved to the extended data section. Similarly, Fig 3e and f could be presented in a graphical form and 3d and h could be moved to to extended data as well.
5. Fig 3c: please remove or make clear why there is a gap in the heatmap/hierarchical clustering.
6. Fig. 7 the heatmaps presented are quite extensive. Could a core set of data be presented in a more concise way and the large heatmaps be moved to the extended data section?
7. Please add densitometric quantifications of western blots.
8. According to the data presented in Supp. Fig 5 the OLFM related signature does not correspond with the developmental trajectory of human NB and normal adrenal medulla, which seem to contradict the hypothesis of the authors. This is not properly interpreted and discussed.
9. The interpretation of data presented in Supp. Fig 7 is not convincing. There is actually no difference in the expression of the OLFM-related signature in MES vs NOR cells in the data set published by Dong et al (Supp. Fig 7d). Furthermore, the fraction of MES cells in the dataset seems overly large.

Reviewer #2 (Remarks to the Author):

Please shorten the Discussion. There is a lot of background knowledge present in the discussion that could be mentioned in the introduction instead. There are some instances in which wording is not easy to understand.

Reviewer #3 (Remarks to the Author):

The authors have made significant efforts to address the comments. There is just one point that requires clarification:

The analysis of OLFM1-mediated downstream signaling is interesting as is the identification of RTN4R as mediating the dissemination of neuroblastoma cells. The authors show in Fig. 8 that OLFM1 may signal through the RTN4R receptor compared to two other receptors in neuroblastoma cells. However, an important control is whether OLFM1 activates this receptor in sympathoblasts or dorsal ganglion cells. If the premise is that OLFM1 is unique to metastatic cells, then such signaling should not occur in "normal" cells.

REVIEWERS' COMMENTS

Reviewer #1 (Remarks to the Author):

Comments to Dounia Ben Amar et al – revised manuscript

The authors now provide new data and have substantially revised the manuscript in response to the concerns and comments raised by me and the other reviewers. The manuscript has substantially improved, but I have still concerns regarding the interpretation of the newly added analysis of bulk and single cell RNA-sequencing datasets. Furthermore, the manuscript needs some further revision, especially the abstract and results section and it is recommended to condense the figures in order to present a coherent and concise storyline.

1. The title of the manuscript shall make clear that embryonic neural crest derivatives is referring to cells.

We tried to make the title as explicit and complete as possible in 15 words.

2. The abstract should be revised to present the main findings in a clear way.

The abstract has now been modified according to the editorial recommendations.

3. The language and specific terms used are sometimes not entirely clear and accurate. Please revise the manuscript throughout with a focus on precise and concise language.

The manuscript has been revised to improve the spelling and has been made much more concise. We hope these modifications will facilitate the reading.

4. While the effort the authors put into presenting additional new data in Figure 7 and other figures is appreciated as they provide new insights and improve the manuscript, the results section is now overly long and should be condensed. It is suggested to reduce the figures to present only the core relevant data as main figures. E.g. Fig 2c does not add much information, Fig. 2e could be presented in a condensed form or moved to the extended data section. Similarly, Fig 3e and f could be presented in a graphical form and 3d and h could be moved to extended data as well.

We now have reduced the main figures as suggested by the reviewer. In particular, Fig 2c and 2e have been moved to a new supplementary figure 3; Fig 3f and Supp Fig 3fg (now Supp Fig 4gh) are now presented in a graphical form and Fig 3d and 3h have been moved to Supp Fig 4c and 4i.

5. Fig 3c: please remove or make clear why there is a gap in the heatmap/hierarchical clustering.

The gap in the heatmap/hierarchical clustering is due to the selected parameters in the heatmap package (version 1.0.12) applied in R software, that computes a distance measure based on a Pearson correlation test to group together genes and/or experimental conditions having highly similar transcriptomic behaviors.

6. Fig. 7 the heatmaps presented are quite extensive. Could a core set of data be presented in a more concise way and the large heatmaps be moved to the extended data section?

We agree with the referee but we cannot make the heatmaps more concise. Indeed, the selection of the genes in the signatures in Fig 7 was based on objective criteria (FC, p-val, enrichment in stage 4 VS stage 1-2-3 neuroblastoma in two cohorts). The selected genes were neither ranked nor chosen for their described functions. Thus, it would not be pertinent to select only a subset of these genes.

7. Please add densitometric quantifications of western blots.

Densitometric quantifications have been added below each blot.

8. According to the data presented in Supp. Fig 5 the OLFM related signature does not correspond with the developmental trajectory of human NB and normal adrenal medulla, which seem to contradict the hypothesis of the authors. This is not properly interpreted and discussed.

We have clarified the interpretation of Supp. Fig 5 (now Supp. Fig 6) in the manuscript. Our intention was not to hypothesize that the OLFM-related signature could follow the developmental trajectory of normal adrenal medulla and subsequently that of human NB. Rather, Supp Fig 6 aims at showing that the whole OLFM gene signature is indeed expressed throughout the physiological lineage, but with a high degree of heterogeneity among cell identities, and

particularly within the “neuroblasts branches”. Our hypothesis is that the different responses of NB patient samples to OLFM1 blocking antibody in the graft experiments could take its root in the heterogeneous expression of OLFM-related partners in the cell(s) of origin of these NBs.

9. The interpretation of data presented in Supp. Fig 7 is not convincing. There is actually no difference in the expression of the OLFM-related signature in MES vs NOR cells in the data set published by Dong et al (Supp. Fig 7d). Furthermore, the fraction of MES cells in the dataset seems overly large.

In Supp. Fig. 7, this is not the OLFM-related signature that is analyzed but the “Upregulated primary tumor cell escape” signature. For this signature, there is a significant difference in expression between MES vs NOR cells in the Dong’s dataset (Supp. Fig 8g). In the first revised version, we scored the MES/NOR identity of tumor cells based on restricted sets of MES/NOR genes (detailed in Supp. Data 1), which could explain the quite high fraction of MES cells (26%, close to data presented in Dong et al with the signature taken from Boeva et al). In the second revised version, we added the same type of analyses using extended MES/NOR signatures (detailed in Supp. Data 1), which lead to the identification of 1% MES cells (close to data presented in Dong with the signature taken from Van Groningen et al; New Supp Fig 8e and g, lower panels). Even with this more “drastic” scoring, the expression of the “Upregulated primary tumor cell escape” signature remains highly significantly different between MES and NOR cells.

Reviewer #2 (Remarks to the Author):

Please shorten the Discussion. There is a lot of background knowledge present in the discussion that could be mentioned in the introduction instead. There are some instances in which wording is not easy to understand. To shorten the discussion, we removed several paragraphs, especially concerning redundant or too detailed background from the literature on OLFM signaling.

Reviewer #3 (Remarks to the Author):

The authors have made significant efforts to address the comments. There is just one point that requires clarification: The analysis of OLFM1-mediated downstream signaling is interesting as is the identification of RTN4R as mediating the dissemination of neuroblastoma cells. The authors show in Fig. 8 that OLFM1 may signal through the RTN4R receptor compared to two other receptors in neuroblastoma cells. However, an important control is whether OLFM1 activates this receptor in sympathoblasts or dorsal ganglion cells. If the premise is that OLFM1 is unique to metastatic cells, then such signaling should not occur in “normal” cells.

We agree with the referee that characterizing the functionality of OLFM1/RTN4R signaling in sympathoblasts / DRG cells would be interesting. However, we could show here that, although expressed within SGs/DRGs, OLFM1 protein has no impact on sympathoblasts cell-cell cohesion or on global SG development (Fig 7 and Supp Fig 7). Moreover, (1) the potential mode of action of OLFM1 on sympathoblasts has never been investigated, and (2) the downstream signaling partners involved in OLFM1/RTN4R signaling are still poorly characterized and may highly depend on the cellular context. Thus, the read out to follow a potential OLFM1/RTN4R signaling in sympathoblasts would be highly uncertain and would require a dedicated project that is beyond the scope and the conclusions of our study.

We have tried to make it very clear that we do not claim that OLFM1 has no function on sympathoblasts, but that the OLFM1-mediated loss of cell-cell cohesion that we report here is specifically observed for tumoral cells but not for sympathoblasts.